# Pentoxifylline-induced protein expression change in RAW 264.7 cells as determined by immunoprecipitation-based high performance liquid chromatography

Mi Hyun Seo[1], Dae Won Kim[2], Yeon Sook Kim[3], Suk Keun Lee[4,5]*

1 Department of Oral and Maxillofacial Surgery, College of Dentistry, Seoul National University, Seoul, South Korea, 2 Department of Oral Biochemistry, College of Dentistry, Gangneung-Wonju National University, Gangneung, Korea, 3 Department of Dental Hygiene, College of Health & Medical Sciences, Cheongju University, Cheongju, South Korea, 4 Department of Oral Pathology, College of Dentistry, Gangneung-Wonju National University, Gangneung, South Korea, 5 Institute of Hydrogen Magnetic Reaction Gene Regulation, Dae Jeon, South Korea

* sukkeunlee2@naver.com

## Abstract

Although pentoxifylline (PTX) was identified as a competitive non-selective phosphodiesterase inhibitor, its pharmacological effect has not been clearly elucidated. The present study explored the effect of low dose 10 μg/mL PTX (therapeutic dose) compared to high dose 300 μg/mL PTX (experimental dose) in RAW 264.7 cells through immunoprecipitation-based high performance liquid chromatography (IP-HPLC), immunohistochemistry, and western blot. 10 μg/mL PTX increased the expression of proliferation (Ki-67, PCNA, cyclin D2, cdc25A), epigenetic modification (KDM4D, PCAF, HMGB1), protein translation (DOHH, DHPS, eIF5A1), RAS signaling (KRAS, pAKT1/2/3, PI3K), NFkB signaling (NFkB, GADD45, p38), protection (HSP70, SOD1, GSTO1/2), survival (pAKT1/2/3, SP1, sirtuin 6), neuromuscular differentiation (NSEγ, myosin-1a, desmin), osteoblastic differentiation (BMP2, RUNX2, osterix), acute inflammation (TNFα, IL-1, CXCR4), innate immunity (β-defensin 1, lactoferrin, TLR-3, -4), cell-mediated immunity (CD4, CD8, CD80), while decreased the expression of ER stress (eIF2α, eIF2AK3, ATF6α), fibrosis (FGF2, CTGF, collagen 3A1), and chronic inflammation (CD68, MMP-2, -3, COX2) versus the untreated controls. The activation of proliferation by 10 μg/mL PTX was also supported by the increase of cMyc-MAX heterodimer and β-catenin-TCF1 complex in double IP-HPLC. 10 μg/mL PTX enhanced FAS-mediated apoptosis but diminished p53-mediated apoptosis, and downregulated many angiogenesis proteins (angiogenin, VEGF-A, and FLT4), but upregulated HIF1α, VEGFR2, and CMG2 reactively. Whereas, 300 μg/mL PTX consistently decreased proliferation, epigenetic modification, RAS and NFkB signaling, neuromuscular and osteoblastic differentiation, but increased apoptosis, ER stress, and fibrosis compared to 10 μg/mL PTX. These data suggest PTX has different biological effect on RWA 264.7 cells depending on the concentration of 10 μg/mL and 300 μg/mL PTX. The low dose 10 μg/mL PTX enhanced RAS/NFkB signaling, proliferation, differentiation, and inflammation, particularly, it stimulated neuromuscular and osteoblastic differentiation, innate immunity, and cell-

**Data Availability Statement:** All relevant data are within the paper and its Supporting Information files.

**Funding:** The authors received no specific funding for this work.

**Competing interests:** The authors have declared that no competing interests exist.

mediated immunity, but attenuated ER stress, fibrosis, angiogenesis, and chronic inflammation, while the high dose 300 μg/mL PTX was found to alleviate the 10 μg/mL PTX-induced biological effects, resulted in the suppression of RAS/NFkB signaling, proliferation, neuro-muscular and osteoblastic differentiation, and inflammation.

## Introduction

Pentoxifylline (PTX), a xanthine derivative, is primarily used as an antiproteolytic agent to treat muscle pain in people with peripheral artery disease [1,2] by activating cAMP/EPAC/AKT signaling [3,4]. It has been frequently reported that PTX remarkably suppressed the secretions of pro-inflammatory cytokines and the nuclear factor-kappa B (NFkB) activation [5–7], and reduced chronic inflammation [5,8]. PTX appeared to have anti-fibrotic effect on radiation-induced lung fibrosis by modulation of PKA and PAI-1 expression as possible anti-fibrotic mechanisms [9], and PTX therapy with vitamin E showed prevention of radiation-induced fibrosis in breast cancer patients [10]. PTX is suggested as an oral osteogenic drug for the treatment of post-menopausal osteoporosis [11]. As PTX given before tooth extraction is prophylactic, it might affect healing in a positive way by optimizing the inflammatory response [12]. Many authors suggest that PTX may increase the anticancer potential of anticancer drugs such as cisplatin or doxorubicin as well as reduce side effects of these drugs [13–16].

As RAW 264.7 cells are immortalized macrophages which are mainly involved with wound healing and tumor progression, the present study utilized RAW 264.7 cells for *in vitro* protein expression experiment. Although PTX has short half-life (0.39–0.84 h for the various doses and 0.96–1.61 h for the metabolites), its therapeutic dose for adult human is usually 400 mg (Trental), three times a day [17,18]. Therefore, in this study, RAW 264.7 cells were primarily treated with 10 μg/mL PTX, which is similar to human therapeutic dose (6.7 mg/kg, Trental). However, in the pilot study to know the trends of protein expressions by PTX, 10 μg/mL PTX increased the expression of some proliferation-related proteins, RAS and NFkB signaling proteins, and even some inflammatory proteins in RAW 264.7 cells. These results were contrary to many reports insisting the anti-proliferative and anti-inflammatory effect of PTX. However, it was found that many experiments for PTX-induced effects on cells and animals were frequently performed by using higher dose of PTX, 100–500 μg/mL [5,8,19–22], than therapeutic dose of PTX, about 10 μg/mL. In order to elucidate the different pharmacological effect depending on the dose of PTX, the present study was performed to compare 10 μg/mL PTX-induced protein expressions with 300 μg/mL PTX-induced protein expression in RAW 264.7 cells.

As the essential protein signalings are intimately correlated and cross-talked with each other to maintain cellular homeostasis during proliferation, differentiation, inflammation, apoptosis, and senescence, it is necessary to know the global protein expression involving multiple signaling pathways in order to explain or predict the fate of cells involved with diseases or drug therapy. The present study examined PTX-induced global protein expression changes in RAW 264.7 cells through IP-HPLC, immunocytochemistry (ICC), and western blot. Particularly, IP-HPLC analysis is available to determine protein expression levels in different biological fluids, such as blood plasma, urine, saliva [23,24], inflammatory exudates [25–27], cancer tissues [28,29], cell culture extract [30–34], and blood plasma [32,35]. Contrary to enzyme-linked immunosorbent assay (ELISA). IP-HPLC uses protein A/G agarose beads in chaotic buffer solution and micro-sensitive UV spectroscopy to determine protein expression level

(%) compared to the control. It can rapidly analyze multiple proteins samples. Furthermore, multiple repeats of same protein IP-HPLC showed accurate results (±5% standard deviation) reproducibly [33,34]. Therefore, 10 μg/mL PTX-induced global protein expression changes in RAW 264.7 cells were extensively examined through IP-HPLC partly with immunocytochemistry (ICC) and western blot methods in this study.

## Materials and methods

### RAW 264.7 cell culture in the presence of PTX

RAW 264.7 cells (an immortalized murine macrophage cell line; ATCC, USA) were cultured in Dulbecco's modified Eagle's medium (WelGene Inc. Korea) supplemented with 10% (vol/vol) heat-inactivated fetal bovine serum (WelGene Inc. Korea), 100 unit/mL penicillin, 100 μg/mL streptomycin, and 250 ng/mL amphotericin B (WelGene Inc. Korea), in 5% $CO_2$ chamber at 37.5˚C. The cells from passage no. 10–12 were used in this study [36]. Antigen free media was utilized in order to detect native protein expressional changes induced by PTX.

About 70% confluent RAW 264.7 cells grown on Petri dish surfaces were treated with 10 μg/mL PTX or 300 μg/mL PTX (safe single dose in dogs 100–300 mg/kg according to WHO food additives Series 35, 835) for 12, 24, or 48 h; control cells were treated with 1 mL of normal saline. Cultured cells were harvested with protein lysis buffer (PRO-PREP$^{TM}$, iNtRON Biotechnology INC, Korea) cooled on ice, and immediately preserved at -70˚C until required.

### Cytological cell counting for the proliferation index

RAW 264.7 cells were cultured on the surfaces of two-well culture slide dishes (SPL, Korea) until they reached 50% confluence, and then treated with 10 μg/mL PTX for 12, 24, or 48 h. The control was treated with normal saline only. The cells on the culture slides were fixed with 10% buffered formalin solution for 1 hour, stained with hematoxylin, and observed by optical microscope (CX43, Olympus, Japan) at x400 magnification. Thirty representative images were digitally captured in each group (DP-73, Olympus Co., Japan), followed by a cell counting using the IMT i-solution program (version 21.1; Martin Microscope, Vancouver, Canada). The results were plotted on a graph.

### Immunocytochemical staining analysis

When approximately 70% confluent RAW 264.7 cells were spread over the surfaces of two-well culture slide dishes, the cells were treated with 10 μg/mL PTX for 12, 24, or 48 h, while the control cells were treated with 100 μL of normal saline. The cells on the culture slides were fixed with 4% paraformaldehyde solution for 20 min, permeabilized with cooled methanol for 10 min at -20˚C, and applied for immunohistochemistry using selected antisera (the same ones used in IP-HPLC, Table 1); Ki-67 for cellular proliferation, KMD4D and PCAF for epigenetic modification, TNFα, IL-6, TLR3, and TLR4 for inflammation, GSTO1/2, LC3, and GADD153 (CHOP) for endoplasmic reticulum stress, PARP-1 and caspase 3 for apoptosis, NSEγ for neural differentiation, MYH2 for muscular differentiation, TGF-β1, RUNX2, OPG, and BMPR2 for osteoblastic differentiation.

Immunocytochemical (ICC) staining was performed using the indirect triple sandwich method on the Vectastatin system (Vector Laboratories, USA), and visualized using a 3-amino-9-ethylcarbazole solution (Santa Cruz Biotechnology, USA) with no counter staining. The results were observed by optical microscope, and their characteristic images were captured (DP-73, Olympus Co., Japan) and illustrated.

**Table 1. Antibodies used in the study.**

| Proteins | No. | Antibodies |
|---|---|---|
| Proliferation-related proteins | 13 | Ki-67*, PCNA*, PLK4*, MPM2*, CDK4*, cyclin D2*, cdc25A*, BRG1*, p14*, p15/16*, p21*, p27*, lamin A/C* |
| cMyc/MAX/MAD network | 6 (3) | cMyc*, MAX*, MAD1*, (p27, CDK4, cyclin D2) |
| p53/Rb/E2F signaling | 7 (3) | p53, E2F1*, Rb1#, p-Rb1*, (cyclin D2, CDK4, p21) |
| Wnt/β-catenin signaling | 10 | Wnt1*, APC*, β-catenin*, snail*, TCF1*, E-cadherin*, MMP9, vimentin*, VE-cadherin&, AXIN2* |
| Epigenetic modification | 9 | histone H1*, KDM4D$, HDAC10$, MBD4*, DMAP1*, DNMT1*, PCAF*, EZH2*, HMGB1* |
| Protein translation | 8 | DOHH!, DHSP!, eIF5A1!, eIF5A2!, eIF2AK3 (PERK)*, p-eIF2AK3 (Thr981, p-PERK) *, eIF2α^, p-eIF2α (Ser51)^ |
| Growth factor | 23 | TGF-α, TGF-β1$, TGF-β2*, TGF-β3*, SMAD2/3, SMAD4*, p-SMAD4, HGFα*, Met*, FGF1*, FGF2*, FGF7*, GH*, GHRH*, IGF-1*, IGF2R*, HER1*, HER2*, PDGF-A*, CTGF*, ERβ*, ALK1, EGF |
| RAS signaling proteins | 25 | NRAS$, KRAS$, HRAS*, STAT3*, p-STAT3 (p727), pAKT1/2/3, PI3K, Rab1*, RAF-B*, MEKK1, ERK1 (MAPK3)*, p-ERK1 (T202, Y204)$, JNK1 (MAPK8)*, p-JNK1, PKC*, p-PKC1α (Thr514)@, mTOR@, p-mTOR, PTEN, p38 (MAPK14), p-p38 (Thr180, Tyr182), AKAP13, JAK2$, TYK2*, (Thr308), SOS1* |
| NFkB signaling proteins | 23 (10) | NFkB*, IKK*, GADD45*, GADD153 (CHOP)*, p-GADD153, NRF2*, PGC-1α*, AMPKα@, NFAT5, MDR*, SRC1*, NFAT5*, NFATC1*, (p38 (MAPK14), p-p38 (Thr180, Tyr182)*, mTOR@, p-mTOR*, PKC*, p-PKC1α@, JAK2*, RAF-B, MEKK1, AKAP13*) |
| Upregulated inflammatory proteins | 34 | TNFα@, IL-1*, IL-6*, IL-10*, CD4*, CD8*, CD28*, CD31 (PECAM1)*, CD34, CD44 (HCAM)*, CD80 (B7-1) *, MMP1*, MMP1/8*, MMP12$, TIMP1*, TIMP2*, M-CSF*, CXCR4*, CTLA4*, granzyme B, β-defensin-1, MCP-1, HLA-DR, lactoferrin, kininogen*, TLR2*, TLR3*, TLR4*, TLR7*, integrin α2*, integrin α5*, versican*, CRP*, perforin* |
| Downregulated inflammatory proteins | 30 | IL-8*, IL-12*, IL-28*, CD3*, CD20*, CD40*, CD54 (ICAM1)*, CD56 (NCAM) *, CD68*, CD99 (MIC2) *, CD106 (VCAM1) *, MMP2*, MMP3$, MMP9*, MMP10, cathepsin C*, cathepsin G*, cathepsin K*, lysozyme*, hepcidin, α1-antitrypsin&, β-defensin-2, β-defensin-3, COX1*, COX2*, LTA4H&, elafin, integrin β1, LL-37, PD-1 (CD279) |
| p53-mediated apoptosis | 15 (1) | (p53*), p73, MDM2*, BAD*, BAK*, NOXA*, PUMA*, BAX*, BCL2*, APAF1*, caspase 9*, c-caspase 9 *, AIF*, PARP-1*, c-PARP* |
| FAS-mediated apoptosis | 10 | FASL*, FAS*, FADD*, FLIP*, BID*, caspase 3*, c-caspase 3*, caspase 7, c-caspase 8*, c-caspase 10*, |
| Protection- and survival proteins | 30 (7) | HO-1*, HSP27*, HSP70*, HSP90α/β*, SOD1*, GSTO1/2*, NOS1$, TERT*, LC3β*, SVCT2&, SP1@, SP3@, leptin*, SIRT1*, SIRT3*, SIRT6*, SIRT7*, FOXP3*, FOXP4*, PLCβ2*, Klotho*, FOXO1*, FOXO3 *(AMPKα, pAKT1/2/3, mTOR, PKC, p-PKC1α, NRF2, PGC-1α) |
| Endoplasmic reticulum stress proteins | 19 (11) | ATF4*, ATF6α*, BIP*, IRE1α, AP1M1, calnexin, caveolin-1, endothelin-1, (HSP-27, HSP-70, eIF2AK3 * (PERK), p-eIF2AK3 (p-PERK), eIF2α^, p-eIF2α (Ser51)^, GADD153 (CHOP)*, p-GADD153*, PGC-1α*, LC3β*, AIF*) |
| SHH and Notch signaling proteins | 11 (3) | SHH*, PTCH1*, GLI1*, EpCAM*, Notch1*, Jagged2*, HIF1α*, VEGF-A*, (CD44 (HCAM), BCL2, Wnt1) |
| Cytodifferentiation proteins | 28 (9) | α-actin*, p63$, vimentin*, TGase 2+, TGase 4+, HK2*, FAK*, CaM*, CRIP1*, cystatin A*, SOX9, AP1M1*, Krox-25, DLX2, TBX22, laminin α5*, pancreatic lipase, PSA, LGR4 *(caveolin-1*, PKC, p-PKC1α, AKAP13, calnexin, PLC-β2, EpCAM, E-cadherin, VE-cadherin) |
| Neuro-muscular differentiation- | 19 (10) | NSEγ*, NK1R*, GFAP*, S-100*, myosin 1a, MYH2*, desmin*, NF1*, α-SMA*, (CaM, calnexin, CRIP1*, cystatin A, AP1M1*, FAK, SHH, β-catenin, Wnt1, TGase 2) |
| Fibrosis proteins | 19 (16) | (FGF1, FGF2, FGF7, TGF-β1, CTGF, PDGF-A, collagen 3A1, collagen 4, collagen 5A, laminin α5, integrin α2, integrin α5, integrin β1, α1-antitrypsin&, elafin, endothelin-1*), CMG2, PAI-1*, plasminogen* |
| Oncogenesis proteins | 18 (5) | PTEN&, BRCA1&, BRCA2&, ATM*, maspin*, DMBT1*, PIM-1*, CEA$, 14-3-3θ*, survivin@, mucin 1*, mucin 4*, YAP1*, (NF1*, PTCH-1, MBD4, p53, Rb1) |
| Angiogenesis proteins | 26 (12) | HIF1α&, angiogenin$, VEGF-A*, VEGF-C*, VEGF-D*, VEGFR2*, p-VEGFR2 (Y951), vWF$, CMG2$, FLT4$, LYVE1*, fibrinogen*, kininogen-1, TEM8, (plasminogen*, FGF2, PDGF-A, CD31 (PECAM1)*, CD44 (HCAM)*, CD54 (ICAM-1)*, CD56 (NCAM), CD106 (VCAM1), MMP2, MMP10, PAI-1*, endothelin-1*) |
| Osteogenesis proteins | 23 (7) | BMP2*, BMP3*, BMP4*, BMPR1B, BMPR2, osteocalcin*, osteopontin*, osteonectin*, RUNX2*, osterix*, ALP*, aggrecan*, OPG*, RANKL*, DMP1*, SOSTDC1*, (versican*, TGF-β1, CTGF, cathepsin K*, HSP-90α/β*, SMAD4, ATF4) |
| Control housekeeping proteins | 3 | α-tubulin*, β-actin*, GAPDH* |

*(Continued)*

**Table 1.** (Continued)

| Proteins | No. | Antibodies |
|---|---|---|
| Total | 409 (97) | |

* Santa Cruz Biotechnology, USA

# DAKO, Denmark

$ Neomarkers, CA, USA

@ ZYMED, CA, USA

&Abcam, Cambridge, UK

^Cell signaling technology, USA;ᵗ kindly donated from M. H. Park in NIH, USA [37]

+ kindly donated from S. I. Chung in NIH, USA [38,39]; the number of antibodies overlapped; ().

**Abbreviations:** AIF: Apoptosis inducing factor, AKAP13: A-kinase anchoring proteins 13, ALK1: Activin receptor-like kinase 1, ALP: Alkaline phosphatase, AMPK: AMP-activated protein kinase, pAKT: v-akt murine thymoma viral oncogene homolog, p-AKT1/2/3 phosphorylated (p-AKT, Thr 308), APAF1: Apoptotic protease-activating factor 1, APC: Adenomatous polyposis coli, ATF4: Activating transcription factor 4, ATM: Ataxia telangiectasia caused by mutations, AXIN2 (axis inhibition protein 2), BAD: BCL2 associated death promoter, BAK: BCL2 antagonist/killer, BAX: BCL2 associated X, BCL2: B-cell leukemia/lymphoma-2, BID: BH3 interacting-domain death agonist, BIP (GRP 78): Binding immunoglobulin protein, BMP2: Bone morphogenesis protein 2, BMPR1B: Bone morphogenetic protein receptor type-1B, BRCA1: Breast cancer type 1 susceptibility protein, BRG1 (SMARCA4): Transcription activator, c-caspase 3: Cleaved-caspase 3, CaM: Calmodulin, CD3: Cluster of differentiation 3, cdc25A: Cell division cycle 25A, CDK4: Cyclin dependent kinase 4, CEA: Carcinoembryonic antigen, CHOP: C/EBP homologous protein, CMG2: Capillary morphogenesis protein 2 (anthrax toxin receptor 2), COX1: Cyclooxygenase-2, CRP: C-reactive protein, CTGF: Connective tissue growth factor, CTLA4: Cytotoxic T lymphocyte-associatedprotein-4, CXCR4: C-X-C chemokine receptor type 4, DHS: Deoxyhypusine synthase, DLX2, homeobox protein Distal-less (Dlx) family, DMAP1: DNA methyltransferase 1 associated protein, DMBT1: Deleted in malignant brain tumors 1, DNMT1: DNA 5-cytosine methyltransferase 1, DOHH: Deoxyhypusine hydroxylase, E2F1: Transcription factor, eIF2AK3 (PERK): Protein kinase R (PKR)-like endoplasmic reticulum kinase (PERK), elF5A1: Eukaryotic translation initiation factor 5A-1, EpCAM: Epithelial cell adhesion molecule, ERβ: Estrogen receptor beta, ERK1 (extracellular signal-regulated protein kinase 1, MAPK3 (mitogen-activated protein kinase 3)), EZH2 (ENX-1): Enhancer of zeste homolog 2, FADD: FAS associated via death domain, FAK: Focal adhesion kinase, FAS: CD95/Apo1, FASL: FAS ligand, FGF1: Fibroblast growth factor 1, FLIP: FLICE-like inhibitory protein, FLT4: Fms-related tyrosine kinase 4, FOXO1 (Forkhead/winged helix box gene, group O 1), FOXP3: Forkhead box P3, GADD45: Growth arrest and DNA-damage-inducible 45, GAPDH: Glyceraldehyde 3-phosphate dehydrogenase, GFAP: Glial fibrillary acidic protein, GH: Growth hormone, GHRH: Growth hormone-releasing hormone, GSTO1/2: Glutathione S-transferase ω 1/2, HCAM (CD44): Homing cell adhesion molecule, HDAC10: Histone deacetylase 10, HER1: Human epidermal growth factor receptor 1, HGFα: Hepatocyte growth factor α, HIF1α: Hypoxia inducible factor-1α, HMGB1 (High Mobility Group box 1 Protein), HO-1: Heme oxygenase 1, HRAS: GTPase HRas, HSP70: Heat shock protein 70, ICAM-1 (CD54): Intercellular adhesion molecule 1, IGF-1: Insulin-like growth factor 1, IGFIIR: Insulin-like growth factor 2 receptor, IKK: Ikappa B kinase, IL-1: Interleukin-1, IRE1α (ERN1): Inositol-requiring enzyme 1 α, JAK2: Janus kinase 2, JNK1: Jun N-terminal protein kinase 1, KDM4D: Lysine-specific demethylase 4D, KRAS: V-Ki-ras2 Kirsten rat sarcoma viral oncogene homolog, LC3β: microtubule-associated protein 1A/1B-light chain 3β, LTA4H: Leukotriene A4 hydrolase, LGR4 (Leucine-rich repeat-containing G-protein coupled receptor 4), LYVE1: Lymphatic vessel endothelial hyaluronan receptor 1, MAD1: Mitotic arrest deficient 1, maspin: Mammary serine protease inhibitor, MAX: Myc-associated factor X, MBD4: Methyl-CpG-binding domain protein 4, MCP1: Monocyte chemotactic protein 1, M-CSF: Macrophage colony-stimulating factor, MDM2: Mouse double minute 2 homolog, MDR: Multiple drug resistance, MEKK1, MMP1: Matrix metalloprotease 1, MPM2: Mitotic protein monoclonal 2, mTOR: Mammalian target of rapamycin, cMyc: V-myc myelocytomatosis viral oncogene homolog, MYH2: Myosin-2, NFkB: Nuclear factor kappa-light-chain-enhancer of activated B cells, NCAM (CD56): Neural cell adhesion molecule 1, NF1: Neurofibromin 1, NFAT5: Nuclear factor of activated T cells 5, NFATC1: Nuclear factor of activated T-cells, cytoplasmic 1, NFkB: Nuclear factor kappa-light-chain-enhancer of activated B cells, NOS1: Nitric oxide synthase 1, NOXA: Phorbol-12-myristate-13-acetate-induced protein 1, NRAS: Neuroblastoma RAS Viral Oncogene homolog, NRF2: Nuclear factor (erythroid-derived)-like 2, OPG: Osteoprotegerin, PAI-1: Plasminogen activator inhibitor-1, PARP-1: Poly-ADP ribose polymerase 1, c-PARP-1: Cleaved-PARP-1, PCAF: p300/CBP-associated factor, PCNA: Proliferating cell nuclear antigen, PD-1 (CD279): Programmed cell death protein 1, PDGF-A: Platelet-derived growth factor-A, PECAM-1 (CD31): Platelet endothelial cell adhesion molecule-1, PERK: Protein-like endoplasmic reticulum kinase, PGC-1α: Peroxisome proliferator-activated receptor gamma coactivator 1α, PI3K: Phosphatidylinositol-3-kinase, PIM-1: Proto-oncogene serine/threonine-protein kinase 1, PKC: Protein kinase C, PLCβ2: 1-phosphatidylinositol-4,5-bisphosphate phosphodiesterase β2, PLK4: Polo like kinase 4 or serine/threonine-protein kinase, PSA: Prostate-specific antigen, PTEN: Phosphatase and tensin homolog, PUMA: p53 upregulated modulator of apoptosis, Rab1: RAS-related protein, Rab GTPases, RAF-B: v-Raf murine sarcoma viral oncogene homolog B, RANKL: Receptor activator of nuclear factor kappa-B ligand, Rb-1: Retinoblastoma-1, RUNX2: Runt-related transcription factor-2, SHH: Sonic hedgehog, SIRT3: Sirtuin 3 (silent mating type information regulation 2 homolog 3), NAD-dependent deacetylase, α-SMA: Alpha-smooth muscle actin, SMAD4: Mothers against decapentaplegic, drosophila homolog 4, SOD1: Superoxide dismutase-1, SOS1: Son of sevenless homolog 1, SOSTDC1: Sclerostin domain-containing protein 1, SOX9: SRY (sex-determining region Y)-related HMG-box transcription factor 9, SP1: Specificity protein 1, SRC1: Steroid receptor coactivator-1, STAT3: Signal transducer and activator of transcription-3, SVCT2: Sodium-dependent vitamin C transporter 2, TBX22: T-box transcription factor 22, TEM8: Tumor-specific endothelial marker 8 (anthrax toxin receptor 1), TERT: Human telomerase reverse transcriptase, TGase 2: Transglutaminase 2, TGF-β1: Transforming growth factor-β1, TNFα: Tumor necrosis factor-α, VCAM-1: Vascular cell adhesion molecule-1, VE-cadherin: Vascular endothelial cadherin, VEGF-A vascular endothelial growth factor A, VEGFR2: Vascular endothelial growth factor receptor 2, p-VEGFR2: Phosphorylated vascular endothelial growth factor receptor 2 (Y951), vWF: Von Willebrand factor, Wnt1: Proto-oncogene protein Wnt1, YAP1: Yes-associated protein 1.

## Western blot analysis

Some representative antisera were utilized for western blot analysis to assess the 10 μg/mL or 300 μg/mL PTX-induced protein expression in RAW 264.7 cells. Ki-67 was selected for cellular proliferation, p53, Rb1, and E2F1 for p53/Rb/E2F signaling, β-catenin, E-cadherin, VE-cadherin, Wnt1, and TCF1 for Wnt/β-catenin signaling, cMyc, MAX, and MAD1 for cMyc/MAX/MAD network, KDM4D and HDAC10 for epigenetic modification, KRAS, HRAS, NRAS, ERK1, and p-ERK1 for RAS signaling, TNFα, TLR2, and TLR4 for inflammation, eIF2AK3, p-eIF2AK3, ATF4, LC3β, and c-caspase 3 for endoplasmic reticulum stress, TGF-β1, BMP2, RUNX2 for osteogenesis, NSEγ and NF1 for neural differentiation, and MYH2 and desmin for muscular differentiation. These antisera were the same ones used in IP-HPLC (Table 1).

The cells treated with 10 μg/mL and 300 μg/mL PTX for 0, 12, 24, and 48 h were collected with phosphate-buffered saline (PBS) separately, treated with trypsin-ethylene-diamine-tetra-acetic acid (trypsin-EDTA) for one minute, and washed with PBS, and followed by cell lysis with ice-cold RIPA buffer (Sigma Aldrich, USA). The lysates were centrifuged at 12,000 g for 20 min at 4˚ C. The protein concentration of the supernatant was quantified using a Bradford assay (BioRad, USA). Equal amounts (30 μg/lane) of the sample proteins were separated by 8, 10, 15, or 20% sodium dodecyl sulfate-polyacrylamide gel electrophoresis (SDS-PAGE) in Tris-glycine SDS running buffer (25 mM Tris, 0.1% SDS, and 0.2M glycine), and transferred to a nitrocellulose membrane. The membranes were blocked with 5% nonfat dry milk in TBST buffer (25 mM Tris-HCl, 150 mM NaCl, 0.1% Tween 20, pH 7.5) for 1 h. After washing three times with TBST buffer, the membrane was incubated with each primary antibody (dilution ratio = 1:1000, the same antibody used in IP-HPLC) and horseradish peroxidase-conjugated secondary antibody for 1 h separately. The protein bands were then detected using an enhanced chemiluminescence system (Amersham Pharmacia Biotech, Piscataway, NJ, USA) and digitally imaged using a ChemiDoc XRS system (Bio-Rad Laboratories, Hercules, CA, USA). The level of β-actin expression was used as an internal control to normalize the expression of the target proteins. The size and intensity of protein bands from the cells treated with 10 μg/mL PTX for 0, 12, 24, and 48 h were demonstrated, and the protein bands from 10 μg/mL PTX-treated cells were compared with those from 300 μg/mL PTX-treated cells.

## Immunoprecipitation-based high performance liquid chromatography (IP-HPLC)

Protein A/G agarose columns were separately pre-incubated with each 1 μg antibody for proliferation-related proteins (n = 13), cMyc/MAX/MAD network proteins (n = 6(3)), p53/Rb/E2F signaling proteins (n = 7(3)), Wnt/β-catenin signaling proteins (n = 10), epigenetic modification proteins (n = 9), protein translation proteins (n = 8), growth factors (n = 23), RAS signaling proteins (n = 25), NFkB signaling proteins (n = 23(10)), upregulated inflammatory proteins (n = 34), downregulated inflammatory proteins (n = 30), p53-mediated apoptosis proteins (n = 15(1)), FAS-mediated apoptosis proteins (n = 10), protection and survival proteins (n = 30(7)), endoplasmic reticulum stress proteins (n = 19(11)), SHH and Notch signaling proteins (11(3)), cytodifferentiation proteins (n = 28(9)), neuromuscular differentiation proteins (19(10)), fibrosis proteins (19(16)), oncogenesis proteins (n = 18(5)), angiogenesis proteins (n = 26(11)), osteogenesis proteins (n = 23(7)), and control housekeeping proteins (n = 3) (numbers in parenthesis indicate number of overlapping antibodies, Table 1). Although the immunoprecipitation is unable to define the size-dependent expression of target protein compared to western blot, it collects every protein containing a specific epitope against

antibody. Therefore, the IP-HPLC can detect whole target proteins, precursor and modified ones, similar to enzyme-linked immunosorbent assay (ELISA).

The supernatant of the antibody-incubated column was removed, and followed by immunoprecipitation-based IP-HPLC. Briefly, each protein sample was mixed with 5 mL of binding buffer (150mM NaCl, 10mM Tris pH 7.4, 1mM EDTA, 1mM EGTA, 0.2mM sodium vanadate, 0.2mM PMSF and 0.5% NP-40) and incubated in the antibody-bound protein A/G agarose bead column on a rotating stirrer at room temperature for 1 h. After multiple washing of the columns with Tris-NaCl buffer, pH 7.5, in a graded NaCl concentration (0.15–0.3M), the target proteins were eluted with 300μL of IgG elution buffer (Pierce, USA). The immunoprecipitated proteins were analyzed using a precision HPLC unit (1100 series, Agilent, Santa Clara, CA, USA) equipped with a reverse-phase column and a micro-analytical UV detector system (SG Highteco, Hanam, Korea). Column elution was performed using 0.15M NaCl/20% acetonitrile solution at 0.5 mL/min for 15 min, 40˚C, and the proteins were detected using a UV spectrometer at 280 nm. The control and experimental samples were run sequentially to allow comparisons. For IP-HPLC, the whole protein peak areas (mAU*s) were obtained and calculated mathematically using an analytical algorithm (see S1 Fig) by subtracting the negative control antibody peak areas, and protein expression levels were compared and normalized using the square roots of protein peak areas. The ratios of the protein levels between the experimental and control groups were plotted into line and star graphs. Protein expressional changes of less than ±5%, ±5–10%, ±10–20%, or over ±20% changes were described as minimal, slight, significant, or marked, respectively [30–33,40]. The housekeeping proteins including $\beta$-actin, $\alpha$-tubulin, and glyceraldehyde 3-phosphate dehydrogenase (GAPDH) were simultaneously used as internal controls.

In the previous study, the IP-HPLC results were compared with western blot data using cytoplasmic housekeeping protein (β-actin), the former showed minute error ranges less than ± 5% which were appropriate for statistical analysis, while the latter showed a large error range of more than 20% which were impossible to be analyzed statistically [40] (see S2 Fig). Therefore, the present study mainly performed IP-HPLC, and its results were compared to representative findings of ICC and western blot performed with some selected antisera, even though ICC and western blot are usually involved with great error range (≥20%).

## Double IP-HPLC

The double IP-HPLC was designed to detect a protein complex or a binding body contained two different proteins. The first IP-HPLC was performed using the first antibody against one protein of complex as the above procedures to get 300 μL protein elute, which was applied as a protein sample in the second IP-HPLC. Subsequently, the second IP-HPLC was performed using the second column containing protein A/G beads bound with the second antibody against the other protein of complex. The protein elute sample was incubated with the protein A/G beads in the second column, and followed the same procedures of IP-HPLC described above.

## Global protein expression indexes

As the overexpression and under-expression of essential 150 proteins observed in this study showed characteristics of 21 cellular functions affected by 10 μg/mL PTX. The maximum expression value (%) of upregulated proteins and the minimum protein expression values (%) of downregulated proteins at 12, 24, 48 h after 10 μg/mL PTX treatment were selected and plotted into a star graph.

## Statistical analysis

Proportional data (%) of the experimental and control groups were plotted on line graphs and star plots. Their analyses were repeated two to six times until the standard deviations reached $\leq\pm5\%$. The line graphs revealed time-dependent expression changes between the relevant proteins, and the star plots revealed the different expression levels of all proteins examined. The results were analyzed by measuring the standard error ($s = \pm\sqrt{\frac{\sigma^2}{n}}$). The expression of the control housekeeping proteins, i.e., $\beta$-actin, $\alpha$-tubulin, and glyceraldehyde 3-phosphate dehydrogenase (GAPDH) was non-responsive ($\leq5\%$) to 12, 24, or 48 h of PTX treatment [40,41] (see S1 File).

## Results

### Proliferation by cytological cell counting assay

10 μg/mL PTX-treated RAW 264.7 cells were evenly spread on two-well culture slide dishes and cultured for 48 h. Their monotonous small round nuclei were stained with hematoxylin, and then well distinguishable under microscope. They were increased in cell number depending on time, at 0, 12, 24, and 48 h (Fig 1A–1D). The number of RAW 264.7 cells observed at x400 magnification was 1273.2±72.48 at 0 h, as a control, 1317.1±71.42 at 12 h, 1358±61.16 at 24 h, and 1415±89.41 at 48 h after PTX treatment. 10 μg/mL PTX increased the proliferation index of RW 264.7 cells by 3.5%, 6.7%, and 11.2% compared to the untreated controls at 12, 24, and 48 h, respectively (Fig 1E).

### Immunocytochemical observation

The characteristic protein expressions were observed in RAW 264.7 cells through immunocytochemical (ICC) staining, that is, Ki-67, a marker of proliferation was strongly positive in 10 μg/mL PTX-treated cells at 12, 24, and 48 h compared to the untreated controls, and KDM4D and PCAF, markers of histone demethylation and acetylation, respectively, were strongly positive at 12 and 24 h. For the immunoreaction of inflammatory proteins, 10 μg/mL PTX-treated cells showed stronger positivity of TNFα, TLR3, and TLR4 at 24 and 48 h compared to the untreated controls, and increased positivity of IL-6 at 12, 24, and 48 h (Fig 2).

10 μg/mL PTX-treated cells showed slight increase of immunoreaction for GSTO1/2 (a marker of antioxidant and cellular stress) and LC3β (a marker of autophagosome biogenesis), compared to the untreated controls. Caspase 3, a marker of apoptosis executioner was markedly positive in 10 μg/mL PTX-treated cells at 12, 24, and 48 h, while the immunoreaction of GADD153 (CHOP, a marker of endoplasmic reticulum stress) and PARP-1 (a marker of DNA damage) was almost similar in the experimental and control cells (Fig 3).

10 μg/mL PTX-treated cells showed slight increase of NSEγ immunoreaction at 24 and 48 h compared to the untreated controls, and marked increase of MYH2 (a marker of muscular differentiation) immunoreaction at 12, 24, and 48 h. Regarding the markers of osteoblastic differentiation, RUNX2, OPG, BMPR2, and TGF-β1 were markedly positive in 10 μg/mL PTX-treated cells at 12, 24, and 48 h (Fig 4). These results indicate 10 μg/mL PTX affect RAW 264.7 cells to have a potential for neuro-muscular and osteogenic differentiation.

### Western blot detection for selected proteins

For the proteins relevant to proliferation, 10 μg/mL PTX-treated cells showed stronger bands for markers of cell proliferation (Ki-67), p53/Rb signaling (p53, Rb1, and E2F1), Wnt/β-catenin signaling (Wnt1, β-catenin, TCF1), and guided cell migration (E-cadherin and VE-

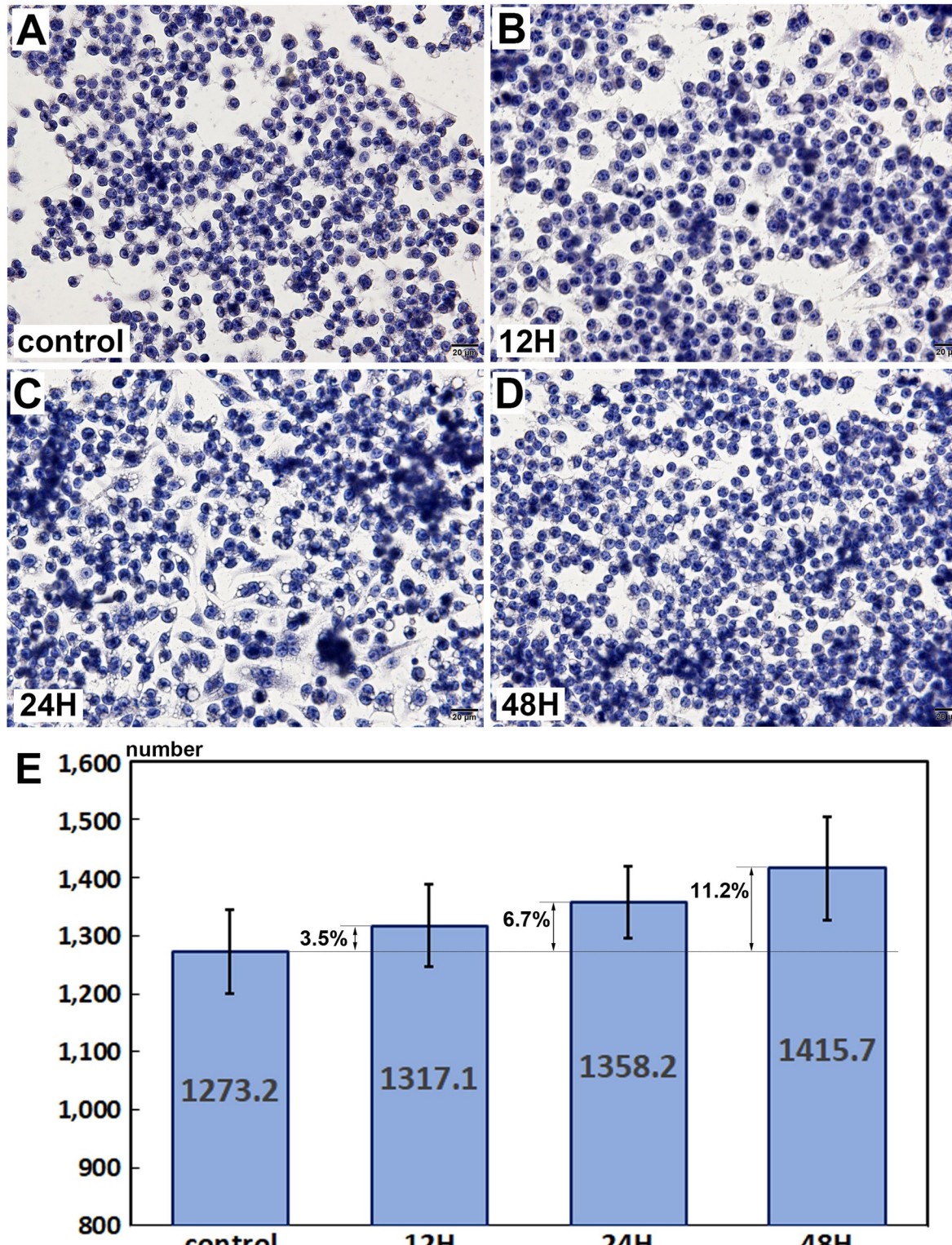

**Fig 1. Proliferation of RAW 264.7 cells by direct cell counting with hematoxylin staining.** 10 μg/mL PTX-treated RAW 264.7 cells showed the increase of proliferation by 3.5% at 12 h, 6.7% at 24 h, and 11.2% at 48 h compared to the non-treated controls. A-D: Histological observation at x400 magnification, E: Statistical analysis plotted into a rod graph, cell number versus culture time (0, 12, 24, and 48 h).

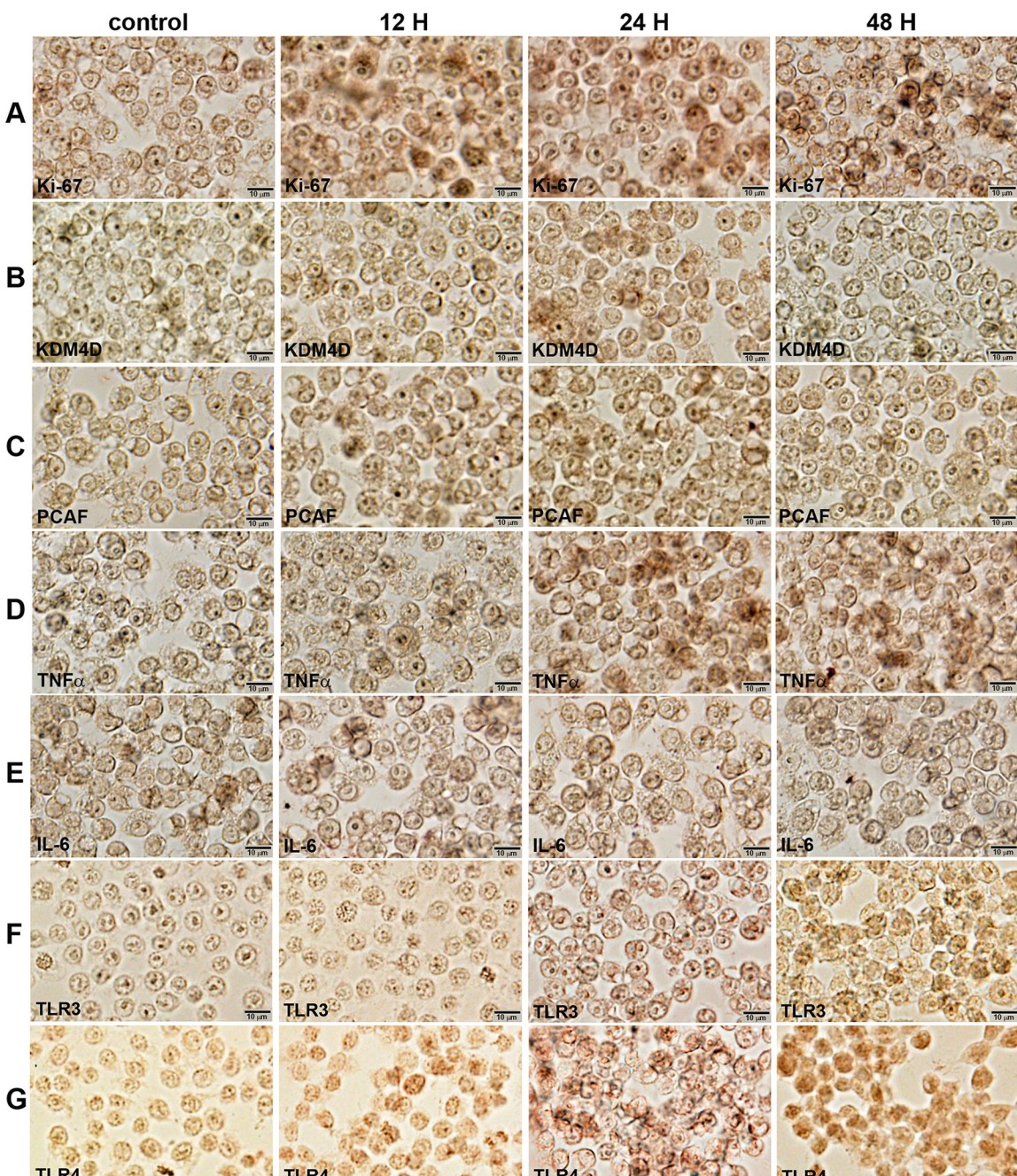

**Fig 2.** Immunocytochemical staining of Ki-67 (A), KMD4D (B), PCAF (C), TNFα (D), IL-6 (E), TLR3 (F), and TLR4 (G) in RAW 264.7 cells after 10 μg/mL PTX treatment for 0, 12, 24, and 48 hours. Noted the cytoplasmic (arrow heads) and nuclear (arrows) localization of different immunoreactions in monocytic round cells. No counter stain.

cadherin) at 12, 24, and 48 h than the untreated controls. Among the cMyc/MAX/MAD network proteins, the bands of cMyc and MAX were gradually attenuated at 12, 24, and 48 h, while the bands of MAD1 were increased. The proteins relevant to epigenetic modification, KDM4D and HDAC10 were increased in 10 μg/mL PTX-treated cells at 12, 24, and 48 h versus the untreated controls. RAS signaling proteins, KRAS, HRAS, NRAS, ERK1, and p-ERK1 were also increased at 12, 24, and 48 h (Fig 5).

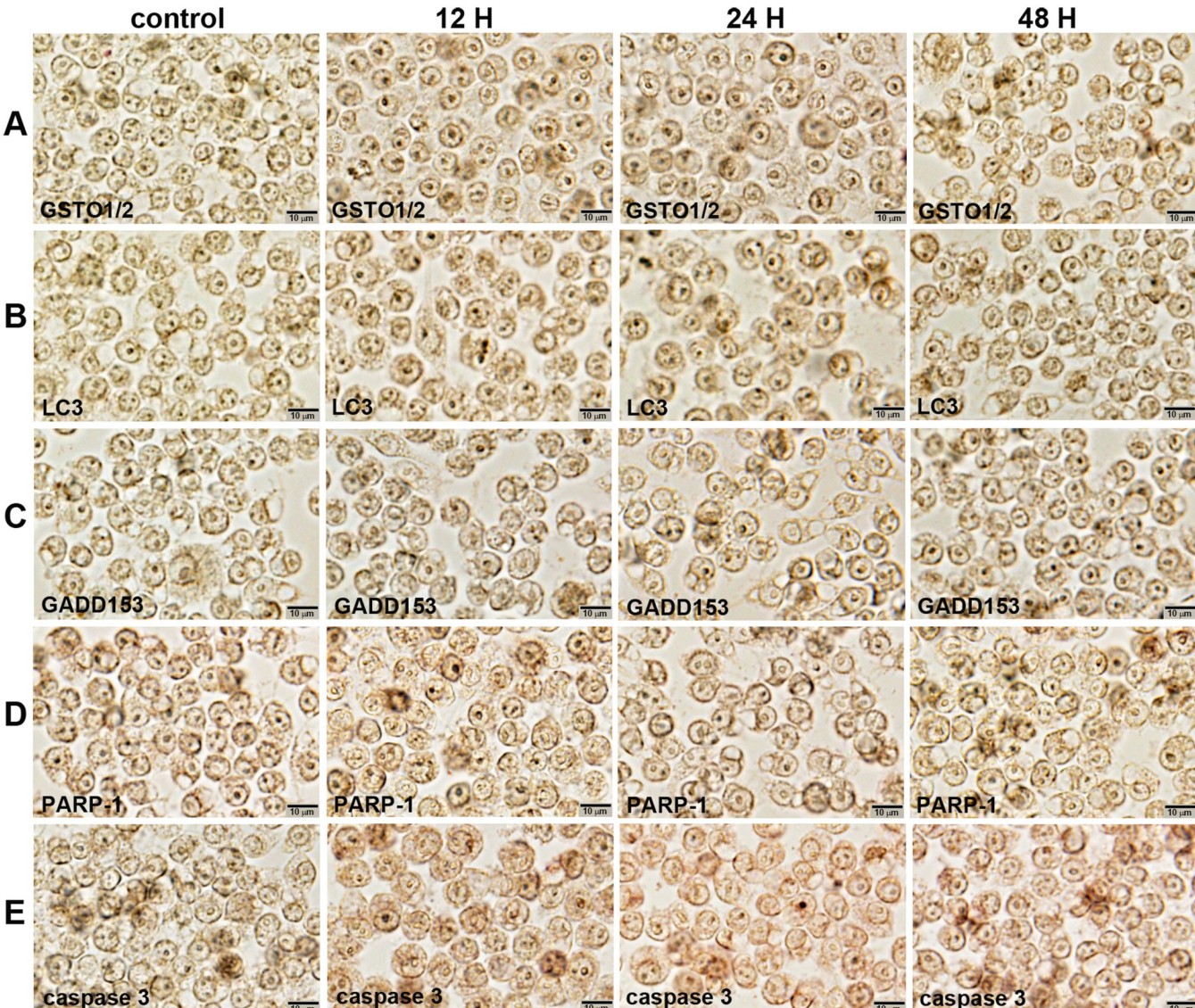

**Fig 3.** Immunocytochemical staining of GSTO1/2 (A), LC3β (B), GADD153 (C), PARP-1(D), and caspase-3 (E) in RAW 264.7 cells after 10 μg/mL PTX treatment for 0, 12, 24, and 48 hours. Noted the cytoplasmic (arrow heads) and nuclear (arrows) localization of different immunoreactions in monocytic round cells. No counter stain.

TNFα, an inflammatory cytokine, TLR2 and TLR4, markers of innate immunity were increased in 10 μg/mL PTX-treated cells at 12, 24, and 48 h compared to the untreated controls. ER stress proteins, eIF2AK3 and p-eIF2AK3, a marker for autophagy formation, LC3β, and an apoptosis executing protein, caspase 3 were coincidently increased at 24 and 48 h. 10 μg/mL PTX-treated cell showed stronger bands of osteoblastic differentiation proteins, TGF-β1, BMP2, RUNX2, and ATF4 than the untreated controls, and slightly strong bands of nerve differentiation proteins, NSEγ and NF1, and muscle differentiation proteins, MYH2 and desmin at 12, 24, and 48 h compared to the untreated controls (Fig 6).

Additionally, 300 μg/mL PTX-induced protein expressions in RAW 264.7 cells were also explored by western blot, and compared with 10 μg/mL PTX-induced protein expressions. 300 μg/mL PTX slightly decreased the expression of Ki-67, cMyc, MAX, E2F1, Wnt1, and

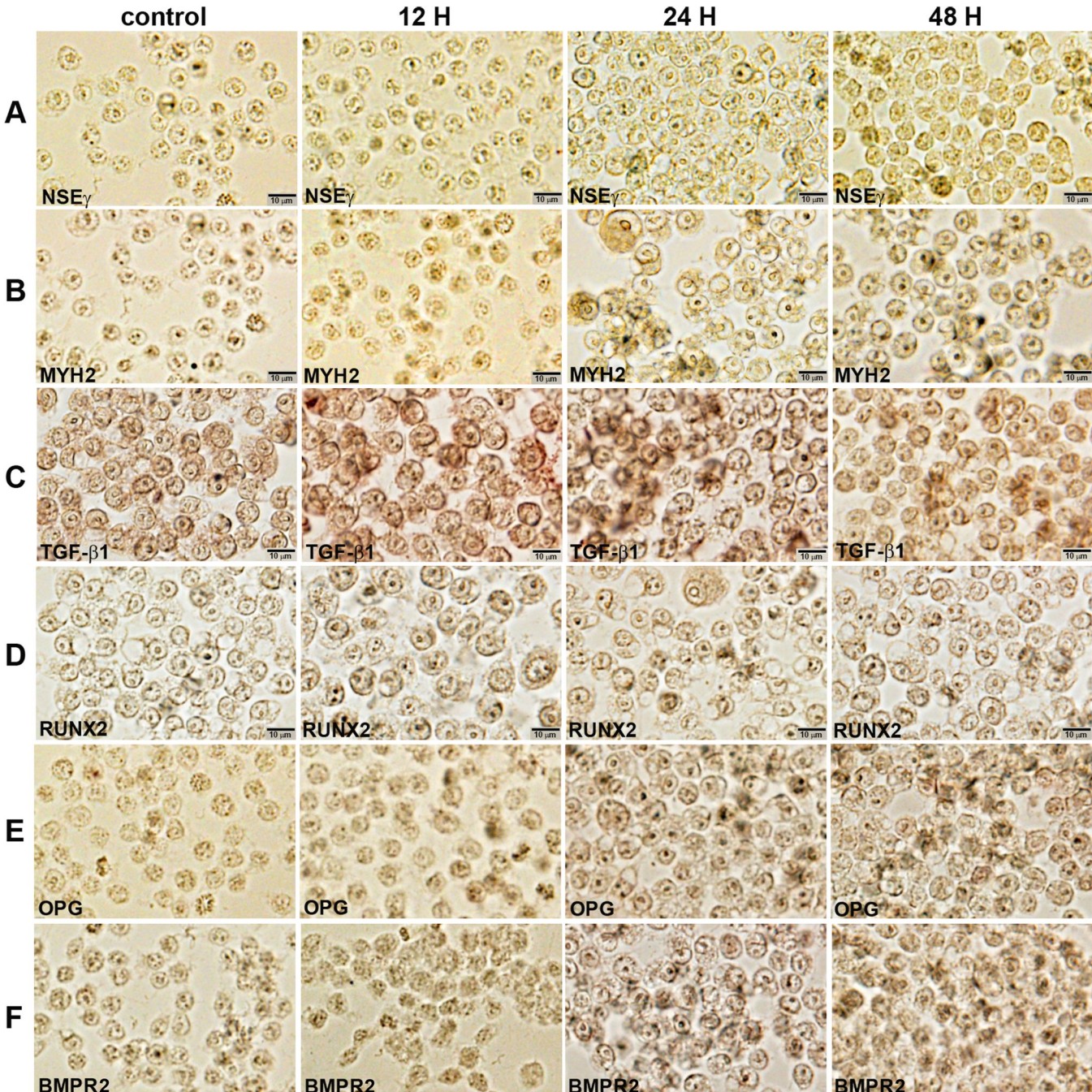

**Fig 4.** Immunocytochemical staining of NSEγ (A), MYH2 (B), TGF-β1 (C), RUNX2 (D), OPG (E), and BMPR2 (F) in RAW 264.7 cells after 10 μg/mL PTX treatment for 0, 12, 24, and 48 hours. Noted the cytoplasmic (arrow heads) and nuclear (arrows) localization of different immunoreactions in monocytic round cells. No counter stain.

TCF1 at 12, 24, and 48 h versus the untreated controls, while 10 μg/mL PTX increased the protein expressions of Ki-67, E2F1, Wnt1, and TCF1, and decreased the protein expressions of cMyc and MAX. The expression of HDAC10 was gradually decreased by 300 μg/mL PTX at 12, 24, and 48 h, while slightly increased by 10 μg/mL PTX (Fig 7).

Regarding RAS signaling, KRAS, ERK1, and pERK1 were slightly downregulated by 300 μg/mL PTX at 12, 24, and 48 h, while upregulated by 10 μg/mL PTX. Apoptosis proteins,

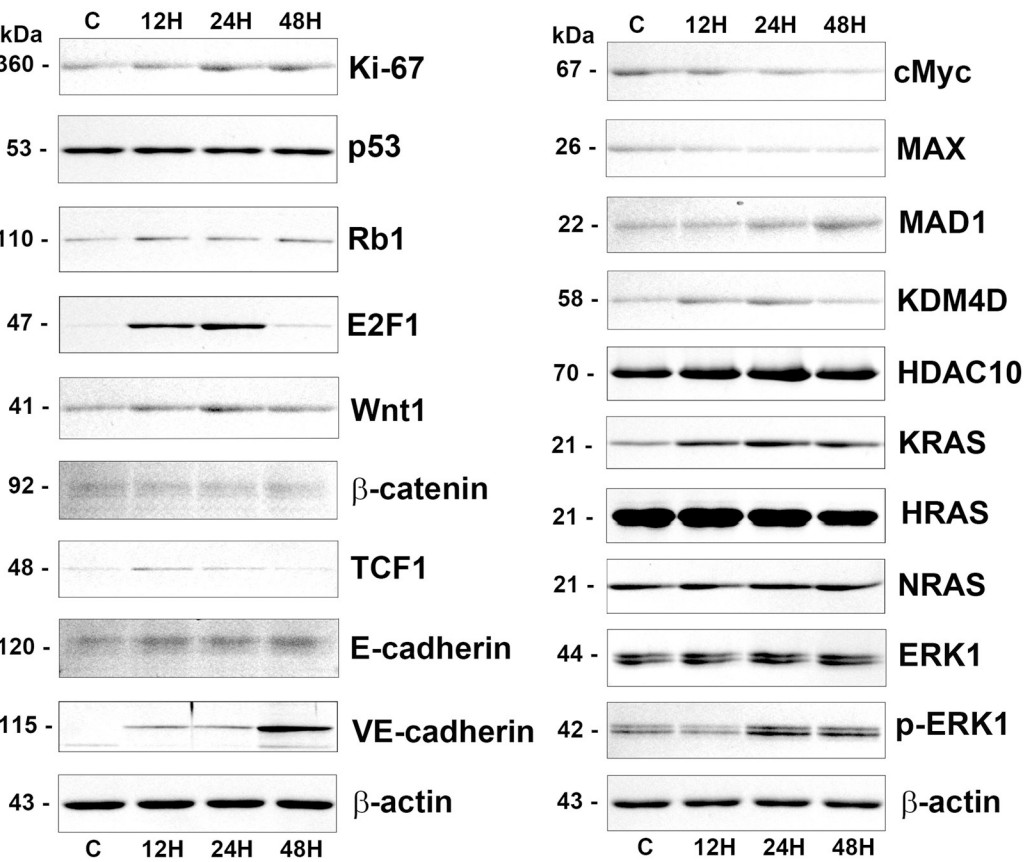

**Fig 5. Western blot analysis for 10 μg/mL PTX-induced protein expression in RAW 264.7 cells regarding the proliferation (Ki-67), p53/Rb/E2F signaling (p53, Rb1, and E2F1), Wnt/β-catenin signaling (Wnt1, β-catenin, and TCF1), guided cell migration (E-cadherin and VE-cadherin), cMyc/MAX/MAD network (cMyc, MAX, and MAD1), epigenetic modification (KDM4D and HDAC10), and RAS signaling (KRAS, HRAS, NRAS, ERK1, and p-ERK1).** The level of β-actin expression was used as an internal control.

p53 and c-caspase 3 were rarely affected by 300 μg/mL PTX, while 10 μg/mL PTX slightly decreased the p53 expression at 12, 24, and 48 h but increased the c-caspase 3 expression at 12, 24, and 48 h. The ER stress proteins, eIF2AK3 and p-eIF2AK3 were slightly upregulated by 300 μg/mL PTX, while eIF2AK3 and p-eIF2AK3 were slightly downregulated but ATF4 was upregulated by 10 μg/mL PTX (Fig 7).

The expressions of inflammatory proteins, TNFα and TLR2 were rarely affected by 300 μg/mL PTX compared to the untreated controls, while increased by 10 μg/mL PTX at 12, 24, and 48 h. And the expressions of osteogenesis proteins, BMP2, RUNX2, and ATF4 were slightly decreased by 300 μg/mL PTX, while increased by 10 μg/mL PTX at 12, 24, and 48 h. On the other hand, the TGF-β1 expression was rarely affected by 300 μg/mL PTX, while slightly increased by 10 μg/mL PTX at 12 and 24 h. And the expression of house-keeping protein, β-actin was almost not affected by 10 μg/mL and 300 μg/mL PTX at 12, 24, and 48 h (Fig 7).

## Immunoprecipitation-based high performance liquid chromatography (IP-HPLC) analysis

10 μg/mL PTX-treated RAW 264.7 cells were extensively explored for different protein expression by IP-HPLC using 409 antisera, and 300 μg/mL PTX-treated cells were simply done using

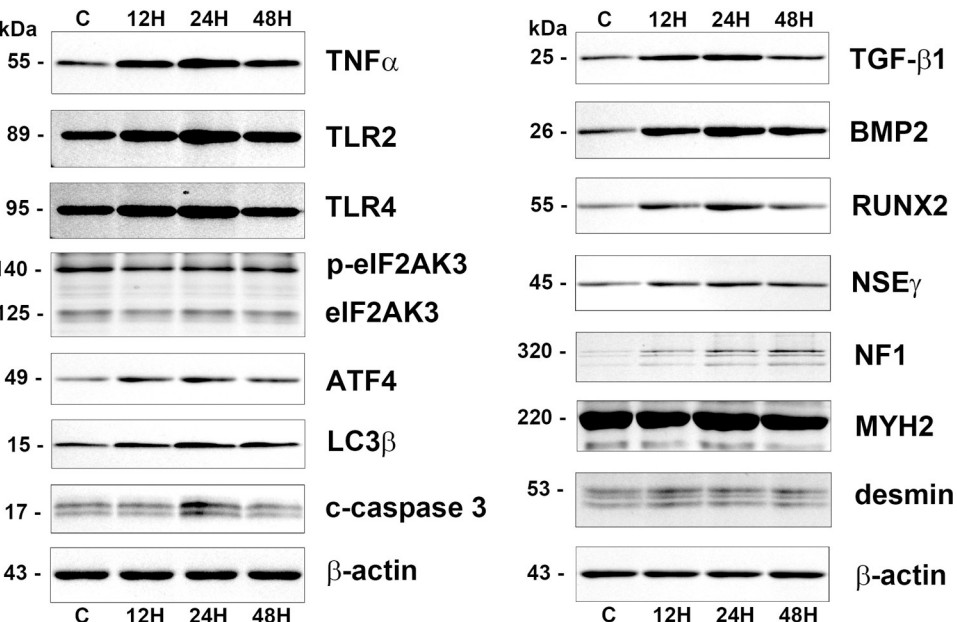

**Fig 6. Western blot analysis for 10 μg/mL PTX-induced protein expression in RAW 264.7 cells regarding the inflammation (TNFα, TLR2, and TLR4), ER stresses (eIF2AK3, p-eIF2AK3, ATF4, LC3β, and caspase 3), osteoblastic differentiation (TGF-β1, BMP2, and RUNX2), neurogenic differentiation (NSEγ and NF1), and muscular differentiation (MYH2 and desmin).** The level of β-actin expression was used as an internal control.

61 antisera. The results of 10 μg/mL PTX-induced protein expression were compared with the results of 300 μg/mL PTX-induced protein expression. The IP-HPLC data were statistically analyzed and illustrated in line and star graphs as follows.

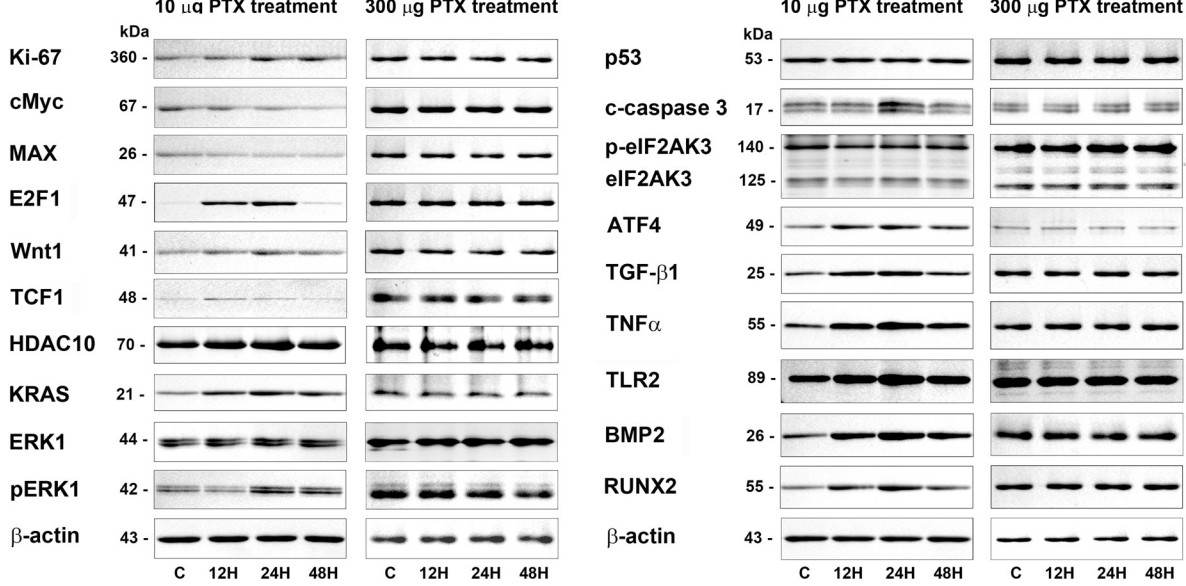

**Fig 7. Western blot comparison between 10 μg/mL and 300 μg/mL PTX-induced protein expressions in RAW 264.7 cells regarding the proliferation (Ki-67, cMyx, MAX, E2F1, Wnt1, and TCF1), epigenetic modification (HDAC10), RAS signaling (KRAS, ERK1, and pERK1), apoptosis (p53 and c-caspase 3), ER stresses (eIF2AK3, p-eIF2AK3, and ATF4), inflammation (TNFα and TLR2), and osteogenesis (BMP2, and RUNX2).** The level of β-actin expression was used as an internal control.

**Effects of 10 μg/mL PTX on the expression of proliferation-related proteins.**   RAW 264.7 cells treated with 10 μg/mL PTX for 12, 24, or 48 h showed significant increases in the expression of proliferation-activating proteins including Ki-67 (by 7.7% at 48h), proliferating cell nuclear antigen (PCNA, 7.3% at 48 h), polo-like kinase 4 (PLK4, a regulator of centriole duplication, 12.1% at 48h), CDK4 (4.2% at 12h), cyclin D2 (a regulator of cyclin-dependent kinase, 10.1% at 12 h), cell division cycle 25A (cdc25A, 6.2% at 12h), transcription factor BRG1 (ATP-dependent chromatin remodeler SMARCA4, 4% at 24h), and reactive increase in the expression of p14ARF (an alternate reading frame (ARF) protein product of the CDKN2A locus, 17.2% at 24 h), p15/16INK (inhibitors of cyclin-dependent kinases (INK), 13% at 48 h), and p21CIP1 (a CDK-interacting protein 1 (CIP1), 5.2% at 48 h) versus the untreated controls. On the other hand, the expressions of mitosis phase promoting factor (MPF) recognized by a mitosis-specific monoclonal antibody 2 (MPM2) and lamin A/C involved in nuclear stability, chromatin structure and gene expression were decreased by 6.7% and 12.1% at 12 h, respectively, and the expression of p27KIP1 (a cyclin dependent kinase inhibitor protein 1 (KIP1)), was minimally affected by PTX (≤5%) (Fig 8A and 8B).

**Effects of 10 μg/mL PTX on the expression of cMyc/MAX/MAD network proteins.** The expression of cMyc (regulator genes and proto-oncogenes that code for transcription factors) was increased by 8.2% at 12h after 10 μg/mL PTX treatment but gradually decreased to the untreated control level at 48 h, the expression of MAX (bHLH-Zip protein forming heterodimer with cMyc) was decreased by 3% at 48 h, while the expression of MAD1 (bHLH-Zip protein forming heterodimer with MAX which can oppose functions of Myc-MAX heterodimers) was increased by 4.8% at 48 h versus the untreated controls (Fig 8C and 8D). Whereas the double IP-HPLC using first antibody of cMyc and second antibody of MAX or MAD1 showed that the heterodimers of cMyc and MAX were increased by 11% at 24 h and 11.9% at 48 h, while the heterodimer of cMyc and MAD1 was decreased by 4.2% at 12 h and 2.1% at 24 h compared to the untreated controls (Fig 8E and 8F). On the other hand, the expressions of cMyc/MAX/MAD network interacting proteins, CDK4 and cyclin D2 were increased by 4.2% at 12 h and 10.1% at 12 h, respectively, but the expression of p27 (cyclin-dependent kinase inhibitor 1B) was minimally affected by PTX (≤5%) (Fig 8C and 8D).

In the double IP-HPLC using antisera of cMyc/MAX and cMyc/MAD1, the cMyc-MAX heterodimer was increased by 11% at 24 h and 11.9% at 48 H, while the cMyc-MAD heterodimer was decreased by 4.2% at 12 h and 2.1% at 24 h compared to the untreated controls. On the other hand, CDK4-p27 complex was consistently reduced by 4.6%, at 12 h, 4.1% at 24 h, and 4.7% at 48h in the double IP-HPLC using CDK4 and p27 antisera (Fig 8E and 8F).

**Effects of 10 μg/mL PTX on the expression of p53/Rb/E2F signaling proteins.**   10 μg/mL PTX decreased the expression of tumor suppressor proteins, that is, p53 by 17.5% at 24 h, Rb1 by 9.8% at 48 h, and phosphorylated Rb1 (p-Rb1) at 5.5% in RAW 264.7 cells versus the untreated controls, while the expression of objective transcription factor, E2F1 was increased by 10.6% at 12 h. And the Ser/Thr-kinase components of cyclin D2 and CDK4 were upregulated by 10.1% and 4.2% at 12 h, respectively, and a cyclin-dependent kinase inhibitor protein, p21CIP1 was also upregulated by 5.2% at 48 h (Fig 9A and 9B).

In the double IP-HPLC using antisera of E2F1/Rb1, CDK4/p21, and E2F1/VE-cadherin, the E2F1-Rb1 and CDK4-p21 complexes were increased by 8.4% and 12.9% at 24 H, respectively, but E2F1-VE-cadherin complex detection as a negative control was minimally affected by 10 μg/mL PTX (Fig 9C and 9D). On the other hand, CDK4-p27 complex was consistently reduced by 4.6% at 12 h, 4.1% at 24 h, and 4.7% at 48h in the double IP-HPLC using CDK4 and p27 antisera (Fig 8E and 8F).

**Effects of 10 μg/mL PTX on the expression of Wnt/β-catenin signaling proteins.**   10 μg/mL PTX increased the protein expressions of Wnt1 (by 13% at 24 h), β-catenin (by 18.5% at

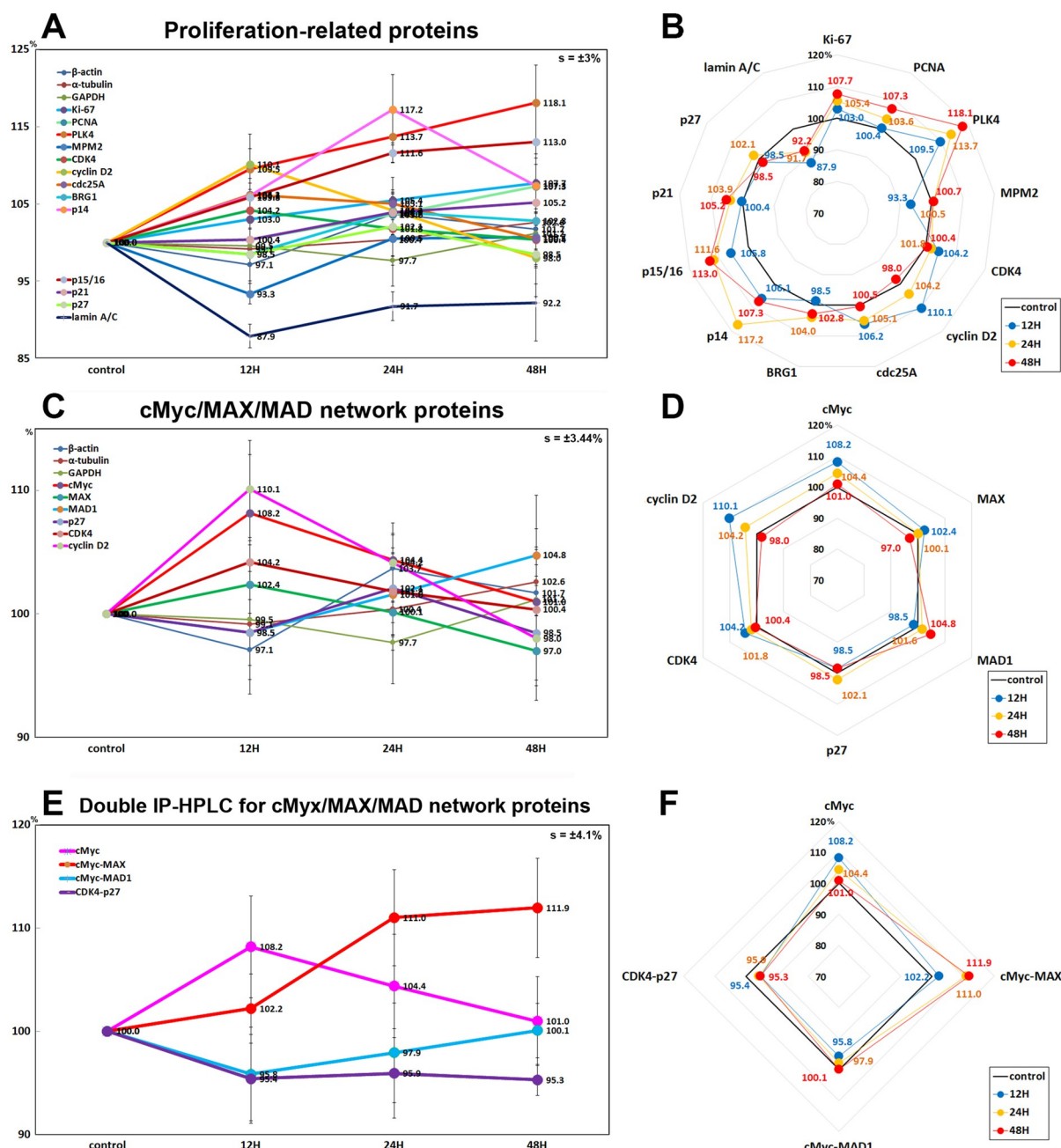

**Fig 8.** Expression of proliferation-related proteins (A and B), cMyc/MAX/MAD network proteins (C and D), and double IP-HPLC for cMyc/MAX/MAD network protein complexes (E and F) in 10 µg/mL PTX-treated RAW 264.7 cells as determined by IP-HPLC. Line graphs, A, C, and E show protein expression on the same scale (%) versus culture time (12, 24, or 48 h), whereas the star plots (B, D, and F) show the differential expression levels of the proteins at 12, 24, or 48 h after PTX treatment on the appropriate scales (%). The thick black line, untreated controls (100%); the blue, yellow, and red dots show differential protein levels after PTX administration for 12, 24, or 48 h, respectively.

24 h), adenomatous polyposis coli (APC, by 13.8% at 48 h), snail (by 13.1% at 48 h), and T-cell factor 1 (TCF1, a transcription factor, by 13.4% at 12 h) versus the untreated controls, while decreased the protein expression of snail (a transcription factor for the repression of adhesion molecule E-cadherin) by 8% at 12 h and AXIN2 (a regulator of β-catenin stability) by 8.7% at

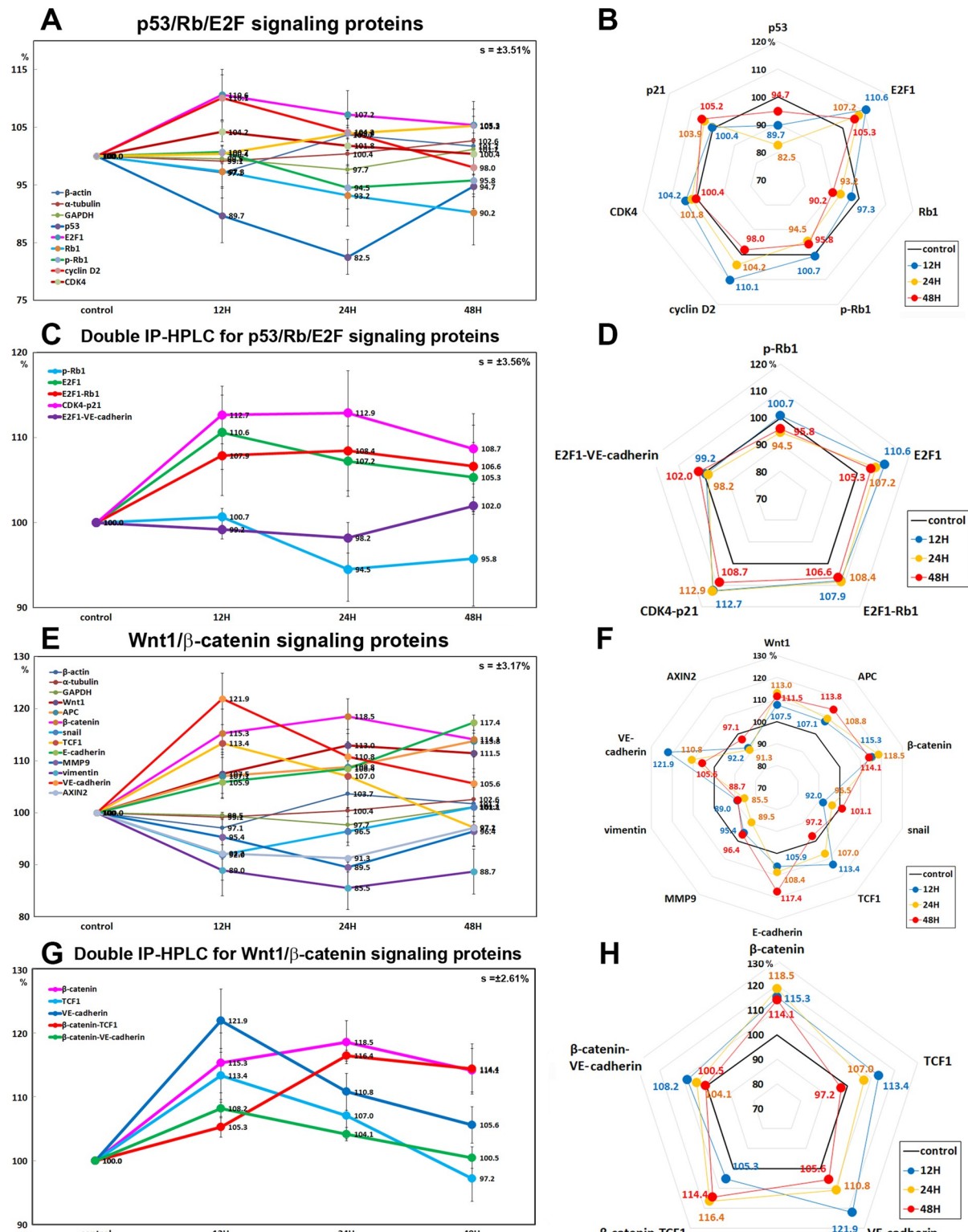

**Fig 9.** Expression of p53/Rb/E2F signaling proteins (A and B), double IP-HPLC for p53/Rb/E2F signaling protein complexes (C and D), Wnt/β-catenin signaling proteins (E and F), and double IP-HPLC for Wnt/β-catenin signaling protein complexes (G and H) in 10 μg/mL PTX-treated RAW 264.7 cells as determined by IP-HPLC. Line graphs, A, C, and E show protein expression on the same scale (%) versus culture time (12, 24, or 48 h), whereas the star plots (B, D, and F) show the differential expression levels of the proteins at 12, 24, or 48 h after PTX treatment on the appropriate scales (%). The thick black line, untreated controls (100%); the blue, yellow, and red dots show differential protein levels after PTX administration for 12, 24, or 48 h, respectively.

24 h. On the other hand, E-cadherin (a type-I transmembrane protein stabilized by β-catenin) and VE-cadherin (vascular endothelial cadherin) were increased by 17.4% at 48 h and 21.9% at 12 h, respectively, while the expressions of Wnt signaling cofactors, MMP9 and vimentin which are necessary in the process of epithelial-mesenchymal transition, were decreased by 10.4% and 12.9% at 24 h, respectively (Fig 9E and 9F).

These results indicate Wnt/β-catenin signaling was enhanced with concomitant upregulation of Wnt1, β-catenin, APC, and TCF1 by 10 μg/mL PTX, but the activation of Wnt/β-catenin signaling was not followed by upregulation of snail, AXIN2, MMP9, and vimentin, but led to the overexpression of E-cadherin and VE-cadherin as shown in the results of double IP-HPLC (Fig 9G and 9H).

**Effects of 10 μg/mL PTX on the expression of epigenetic modification proteins.** 10 μg/mL PTX increased the expression of histone H1 (by 10.6% at 24 h), lysine-specific demethylase 4D (KDM4D, 12% at 48 h), high mobility group box 1 (HMGB1, 8.8% at 24 h), and p300/CBP-associated factor K (lysine) acetyltransferase 2B which has histone acetyl transferase activity (PCAF, by 14% at 12 h) versus the untreated controls, while decreased the expression of histone deacetylase 10 (HDAC10, 9.6% at 24 h), DNA (cytosine-5)-methyltransferase 1 (DNMT1: 23.5% at 48 h), DNA methyltransferase 1-associated protein 1 (DMAP1: 17.7% at 48 h), histone-lysine N-methyltransferase enzyme (enhancer of zeste homolog 2 (EZH2), 6.6% at 24 h), and methyl-CpG binding domain 4 (MBD4: 6.7% at 12 h) (Fig 10A and 10B).

**Effects of 10 μg/mL PTX on the expression of protein translation proteins.** RAW 264.7 cells treated with 10 μg/mL PTX showed increase in the expression of protein translation protein: deoxyhypusine hydroxylase (DOHH, by 7.6% at 12 h), deoxyhypusine protein synthase (DHPS, 17.1% at 24 h), eukaryotic translation initiation factor 5A1 (eIF5A1, 6.5% at 12 h), and eIF5A2 (7.5% at 24 h) versus the untreated controls. On the other hand, the essential factor for protein synthesis to form a ternary complex (TC) with GTP and the initiator Met-tRNA, that is, eIF2α and p-eIF2α were decreased by 8.7% at 12 h and 5.7% at 48 h, respectively, and eukaryotic translation initiation factor 2-α kinase 3 (eIF2AK3, a protein kinase R (PKR)-like endoplasmic reticulum kinase (PERK)) was decreased by 9.3% at 12 h, but p-eIF2AK3 was reactively increased by 6.7% at 48 h (Fig 10C and 10D).

**Effects of 10 μg/mL PTX on the expression of growth factor.** RAW 264.7 cells after 10 μg/mL PTX administration showed marked decrease in the expression of hepatocyte growth factor receptor α (HGFα, by 17.5% at 24 h), Met (HGF receptor, 9.9% at 48 h), fibroblast growth factor-1 (FGF1, 5.5% at 12 h), FGF2 (10.3% at 12 h), FGF7 (2% at 48 h), connective tissue growth factor (CTGF, 17.7% at 24 h), and estrogen receptor-β (ERβ, 14.1% at 24 h) versus the untreated controls. Particularly, PTX decreased the expression of TGF-α (by 21% at 48 h), TGF-β1 (12.5% at 12 h), TGF-β2 (18.5% at 48 h), TGF-β3 (16.4% at 24 h), activin receptor-like kinase 1 (ALK1, 14.6% at 12 h), SMAD2/3 (15.8% at 12 h), and HER1 (epidermal growth factor receptor, 13.4%), while compensatory increased the expression of SMAD4 (14.9% at 48 h), p-SMAD4 (7.1% at 24 h), growth hormone (GH, 11.4% at 48 h), growth hormone releasing hormone (GHRH, 16.9% at48 h), insulin-like growth factor 2 receptor (IGF2R, 11.4% at 48 h), HER2 (EGF receptor tyrosine-protein kinase erbB-2, 9.6% at 24 h), and epidermal growth factor (EGF, 18.8% at 24 h). On the other hand, the expressions of FGF7, IGF1, and PDGF-A were affected minimally by PTX (≤5%) (Fig 10E and 10F).

**Effects of 10 μg/mL PTX on the protein expressions of RAS signaling proteins.** 10 μg/mL PTX affected RAS signaling protein expressions of RAW 264.7 cells positively or negatively. The RAS signaling proteins were initially upregulated at 12 and 24 h and subsequently became similar to control level or downregulated at 48 h by 10 μg/mL PTX, that is, Kirsten Rat Sarcoma virus oncogene (KRAS) to 121% at 12 h and 103.3% at 48 h, neuroblastoma RAS viral oncogene homolog (NRAS) to 102.7% at 24 h and 95.9% at 48 h, GTPase HRAS also known as

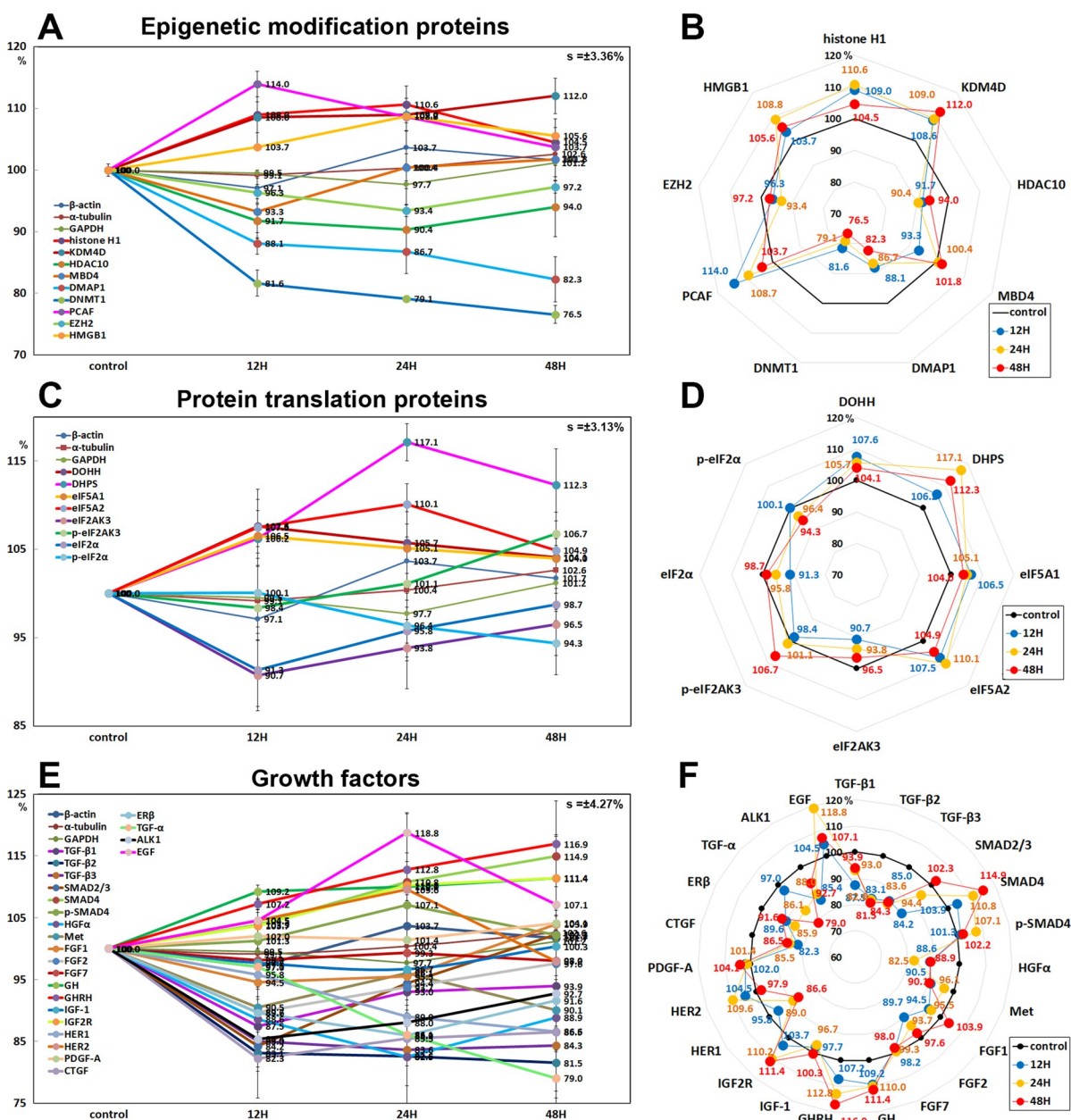

**Fig 10.** Expression of epigenetic modification proteins (A and B), protein translation proteins (C and D), and growth factors (E and F) in 10 μg/mL PTX-treated RAW 264.7 cells as determined by IP-HPLC. Line graphs, A, C, and E show protein expression on the same scale (%) versus culture time (12, 24, or 48 h), whereas the star plots (B, D, and F) show the differential expression levels of the proteins at 12, 24, or 48 h after PTX treatment on the appropriate scales (%). The thick black line, untreated controls (100%); the blue, yellow, and red dots show differential protein levels after PTX administration for 12, 24, or 48 h, respectively.

transforming protein p21 (HRAS) to 110.3% at 12 h and 106.5% at 48 h, signal transducer and activator of transcription 3 (STAT3) to 106.8% at 12 h and 99.1% 48 h, p-STAT3 to 106.7% at 24 h and 99.4% at 48 h, phosphorylated AKT1/2/3 (pAKT1/2/3: Thr 308, a critical mediator of growth factor-induced signals) to 106.5% at 24 h and 102.2% at 48 h, phosphatidylinositol 3-kinase (PI3K) to 113.6% at 24 h and 108% at 48 h, GTPases Rab to 110.2% at 12 h and 101.6% at 48 h, serine/threonine-protein kinase RAF-B to 107.6% at 12 h and 99.3% at 48 h,

non-receptor tyrosine-protein kinase (TYK2, the first member of JAK family) to 122.9% at 12 h and 104.7% at 48 h, protein kinase C (PKC) to 108.6% at 24 h and 99.3% at 48 h, p-PKC1α to 111.5% at 24 h and 103% at 48 h, and A-kinase anchoring proteins (AKAP13) to 116.4% at 24 h and 102.2% at 48 h versus the untreated controls. However, some downstream effector proteins of RAS signaling were consistently upregulated by 10 μg/mL PTX, that is, extracellular signal-regulated kinase 1 (a.k.a. mitogen-activated protein kinase 3, ERK1) by 8.6% at 24 h, pERK-1 by 9.3% at 24 h, p38 by 13.1% at 48 h, and p-p38 by 12.3% at 48 h.

Whereas the expressions of other RAS signaling proteins were decreased, that is, c-Jun N-terminal kinase-1 (JNK1) by 5.6% at 24 h, phosphorylated JNK1 (p-JNK1, Thr 183/Tyr 185) by 11.1% at 24 h, MEK kinase 1 (also designated MAP kinase kinase kinase 1, MKKK1, MAP3K1 or MEKK1) by 11.7% at 12 h, mammalian target of rapamycin (mTOR) by 19.5% at 24 h, phosphorylated mTOR (p-mTOR) by 15.7% at 48 h, Janus kinase 2 (JAK2, non-receptor tyrosine kinase) by 14.8% at 48 h, and son of sevenless homolog 1 (SOS1) by 22.3% at 12 h. On the other hand, the expression of phosphatase and tensin homolog (PTEN) was affected minimally by PTX (≤5%) (Fig 11A and 11B).

**Effects of 10 μg/mL PTX on the expression of NFkB signaling proteins.** 10 μg/mL PTX had different effects on the expression of NFkB signaling proteins in RAW 264.7 cells. PTX markedly upregulated nuclear factor kappa-light-chain-enhancer of activated B cells (NFkB) by 20.7% at 24 h but slightly downregulated ikappa B kinase (IKK) by 6.8% at 12 h versus the untreated controls, and subsequently increased the expression of downstream effector proteins of NFkB signaling, that is, p38 mitogen-activated protein kinase (p38) by 13.1% at 48 h, phosphorylated p38 (p-p38) by 12.3% at 48 h, growth arrest and DNA damage 45 (GADD45) by 18.7% at 48 h, multiple drug resistance (MDR) by 12.5% at 48 h, protein kinase C (PKC) by 8.6% at 24 h, p-PKC1α by 11.5% at 24 h, steroid receptor co-activator-1 (SRC1) by 12.3% at 48 h, and A-kinase anchoring proteins (AKAP13) by 16.4% at 24 h.

On the other hand, the expressions of some downstream regulating proteins of NFkB signaling were decreased, that is, mTOR by 19.5% at 24 h, p-mTOR by 15.7% at 48 h, and peroxisome proliferator-activated receptor gamma coactivator 1-α (PGC1α) by 12.1% at 24 h, MEKK1 by 11.7% at 12 h, JAK2 by 14.8% at 48 h, and nuclear factor of activated T-cells 1 (NFATC1) by 7.3% at 24 h. The expressions of GADD153, nuclear factor (erythroid-derived 2)-like 2 (NRF2), 5' AMP-activated protein kinase α (AMPKα), and nuclear factor of activated T cells (NFAT5) were only minimally affected by PTX (≤ 5%) (Fig 11C and 11D).

**Inflammatory proteins upregulated by 10 μg/mL PTX.** Among the inflammatory proteins, some proteins were upregulated by 10 μg/mL PTX in RAW 264.7 cells as follows, that is, tumor necrosis factor α (TNFα) by 10.6% at 24 h, interleukin-1 (IL-1) by 5.7% at 48 h, IL-6 by 15.2% at 24 h, IL-10 by 10.9% at 24 h, CD4 by 25.3% at 48 h, CD8 by 7.3% at 24 h, CD31 by 4.6% at 48 h, CD34 by 22.2% at 24 h, CD44 (homing cell adhesion molecule (HCAM)) by 9.8% at 48 h, CD80 by 13.3% at 24 h, matrix metalloproteinase 1 (MMP1) by 18.1% at 24 h, MMP1/8 by 10.8% at 48 h, tissue inhibitor of metalloproteinase 1 (TIMP1) by 14.7% at 48 h, TIMP2 by 5.2% at 48 h, macrophage colony-stimulating factor (M-CSF) by 14%% at 48 h, C-X-C chemokine receptor type 4 (CXCR4, a.k.a. CD184) by 13.5% at 48 h, cytotoxic T lymphocyte-associated protein-4 (CTLA4) by 8.4% at 24 h, granzyme B by 9.1% at 12 h but decreased by 10.2% at 12 h, β-defensin 1 by 11.4% at 12 h, monocyte chemotactic protein-1 (MCP1) by 10.1% at 48 h, human leukocyte antigen—DR isotype (HLA-DR) by 13.3% at 48 h, lactoferrin by 16% at 24 h, kininogen by 6.5% at 12 h, toll-like receptor 3 (TLR3) by 10.9% at 48 h, TLR4 by 11.7% at 48 h, integrin α2 by 16% at 24 h, integrin α5 by 12.2% at 48 h, chondroitin sulfate proteoglycan versican by 16.9% at 24 h, and perforin by 14.7% at 24 h. On the other hand, the expression of MMP12, TLR2, TLR7, and C-reactive protein (CRP) showed a trend of increase but were only minimally affected by PTX (≤ 5%) (Fig 11E and 11F).

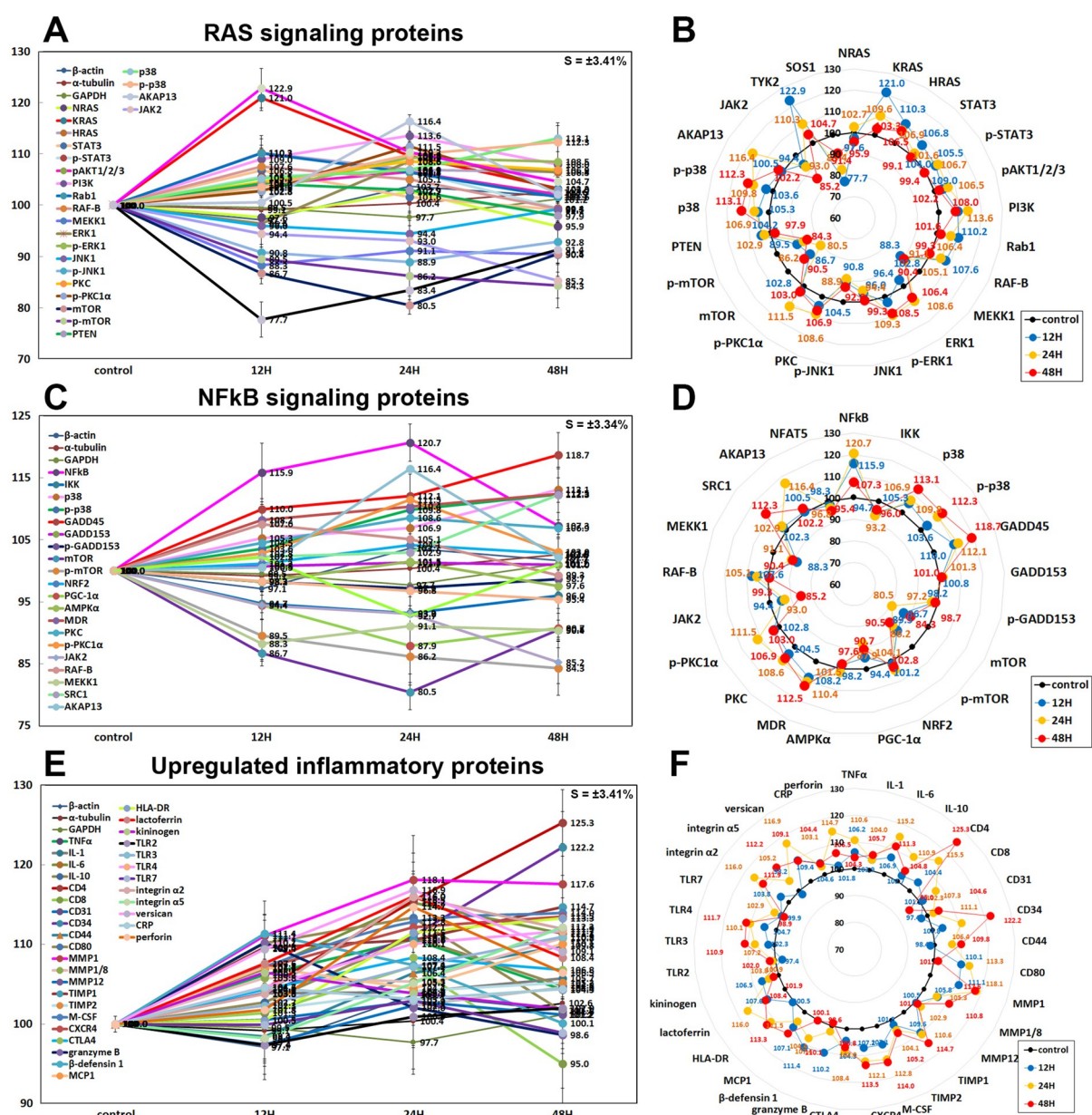

**Fig 11.** Expression of RAS signaling proteins (A and B), NFkB signaling proteins (C and D), and upregulated inflammatory proteins (E and F) in 10 μg/mL PTX-treated RAW 264.7 cells as determined by IP-HPLC. Line graphs, A, C, and E show protein expression on the same scale (%) versus culture time (12, 24, or 48 h), whereas the star plots (B, D, and F) show the differential expression levels of the proteins at 12, 24, or 48 h after PTX treatment on the appropriate scales (%). The thick black line, untreated controls (100%); the blue, yellow, and red dots show differential protein levels after PTX administration for 12, 24, or 48 h, respectively.

**Inflammatory proteins downregulated by 10 μg/mL PTX.** Among the inflammatory proteins, some proteins were downregulated by 10 μg/mL PTX in RAW 264.7 cells as follows: IL-8 by 7.5% at 12 h, IL-12 by 5.4% at 48 h, IL-28 by 17.9% at 48 h, CD3 by 14% at 12 h, CD20 by 18.3% at 12 h, CD28 by 10.2% at 12 h, CD54 (Intercellular Adhesion Molecule 1 (ICAM-1)) by 12.6% at 12 h, CD56 (Neural Cell Adhesion Molecule (NCAM)) by 18.8% at 24 h, CD99 by 15.3% at 12 h, CD106 (Vascular Cell Adhesion Molecule-1 (VCAM-1)) by 15.5% at 12 h, MMP-2 by 16.3% at 24 h, MMP3 by 12.7% at 12 h, MMP9 by 10.5% at 24 h, MMP10 by 9.4%

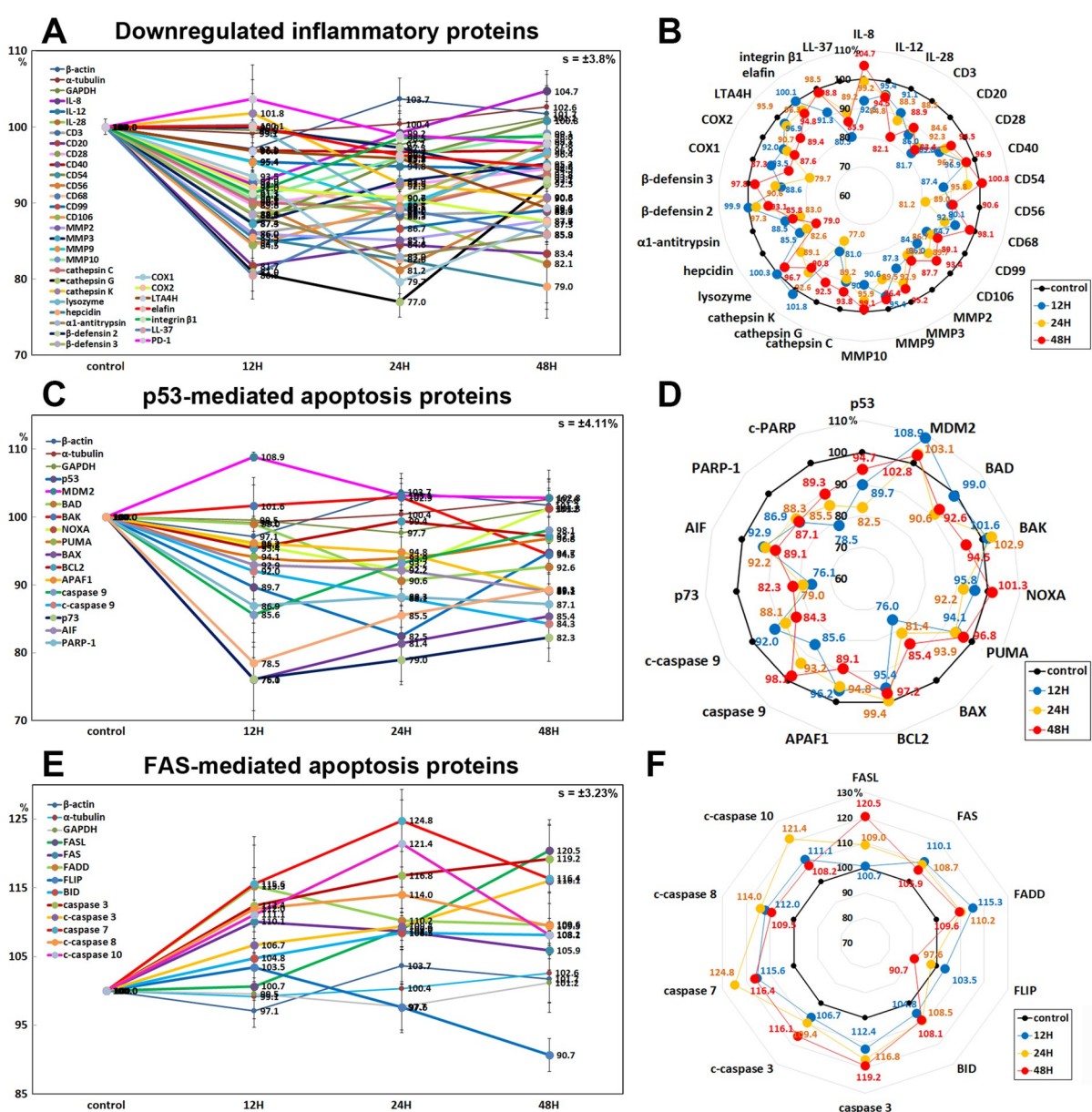

**Fig 12.** Expression of downregulated inflammatory proteins (A and B), p53-mediated apoptosis proteins (C and D), and FAS-mediated apoptosis proteins (E and F) in 10 μg/mL PTX-treated RAW 264.7 cells as determined by IP-HPLC. Line graphs, A, C, and E show protein expression on the same scale (%) versus culture time (12, 24, or 48 h), whereas the star plots (B, D, and F) show the differential expression levels of the proteins at 12, 24, or 48 h after PTX treatment on the appropriate scales (%). The thick black line, untreated controls (100%); the blue, yellow, and red dots show differential protein levels after PTX administration for 12, 24, or 48 h, respectively.

at 12 h, cathepsin C by 10.8% at 24 h, cathepsin G by 23% at 24 h, cathepsin K by 9.2% at 48 h, lysozyme by 10.9% at 24 h, hepcidin by 17.4% at 24 h, α1-antitrypsin by 17% at 24 h, β-defensin 2 by 6.9% at 48 h, β-defensin 3 by 11.4% at 12 h, cyclooxygenase 1 (COX1) by 20.3% at 24 h, COX2 by 12.4% at 48 h, leukotriene A4 hydrolase (LTA4H) by 10.6% at 48 h, elafin by 5.2% at 48 h, integrin β1 by 8.7% at 12 h, and LL-37 by 19.5% at 24 h. On the other hand, the expression of CD40 and programmed cell death protein 1/1 (PD-1, CD279) showed a trend of decrease but were only minimally affected by PTX (≤ 5%) (Fig 12A and 12B).

**Effects of 10 μg/mL PTX on the expression of p53-mediated apoptosis proteins.** 10 μg/mL PTX significantly reduced the expression of p53-mediated pro-apoptotic proteins, that is, p53 by 17.5% at 24 h, BCL2 homologous antagonist/killer (BAK) by 5.5% at 48 h, p73 by 23.9% at 12 h, NOXA (a pro-apoptotic member of the BCL2 protein family) by 7.8% at 24 h, p53 upregulated modulator of apoptosis (PUMA, mitochondria pro-apoptotic BCL-2 homolog) by 6.1% at 24 h, apoptosis regulator BAX by 24% at 48 h, apoptotic protease activating factor 1 (APAF1) by 10.9% at 48 h, caspase 9 by 14.4% at 12 h, and c-caspase 9 by 15.7% at 24 h, apoptosis inducing factor (AIF) by 10.9% at 48 h, poly [ADP-ribose] polymerase 1 (PARP1) by 13.1% at 12 h, and cleaved PARP (c-PARP) by 21.5% at 12 h versus the untreated controls, while it increased the expression of murine double minute-2 homolog (MDM2, negative regulator of p53) by 8.9% at 12 h and minimally affected the expression of B cell lymphoma 2 (BCL2, anti-apoptotic protein) ($\leq$ 5%) (Fig 12C and 12D).

**Effects of 10 μg/mL PTX on the expression of FAS-mediated apoptosis proteins.** RAW 264.7 cells treated with 10 μg/mL PTX showed increases in the expression of FAS-mediated apoptosis proteins, that is, FAS ligand (FASL) by 20.5% at 48 h, FAS (CD95) by 10.1% at 12 h, FAS-associated protein with death domain (FADD) by 15.3% at 12 h, BH3 interacting-domain death agonist (BID) by 8.5% at 24 h, caspase 3 by 19.2% at 48 h, c-caspase 3 by 16.1% at 48 h, caspase 7 by 24.8% at 48 h, c-caspase 8 by 14% at 24 h, c-caspase 10 by 21.4% at 24 h versus the untreated controls, while the expression of FLICE-like inhibitory protein (FLIP) was decreased by 9.3% at 48 h (Fig 12E and 12F).

**Effects of 10 μg/mL PTX on the expression of protection-related proteins.** 10 μg/mL PTX-treated RAW 264.7 cells showed increases in the expression of cellular stress protection- and antioxidant-related proteins versus the untreated controls, as follows: heat shock protein-70 (HSP70) by 17.2% at 12 h, Cu-Zn superoxide dismutase-1 (SOD1) by 28.2% at 48 h, glutathione S-transferase ω 1/2 (GSTO1/2, a detoxifying enzyme) by 30.3% at 48 h, nitric oxide synthases-1 (NOS1) by 6.4% at 12 h, LC3β by 11.6% at 48 h, sodium-dependent vitamin C transporter 2 (SVCT2) by 6.6% at 48 h, a transcription factor regulating the expression of antioxidant proteins NRF2 by 4.1% at 24 h, and energy expenditure hormone leptin by 13.1% at 24 h versus the untreated controls. Whereas PTX decreased the expression of cellular maintenance proteins: heme oxygenase-1 (HO1) by 12.1% at 48 h, small heat shock protein HSP27 by 14.2% at 48 h, HSP90α/β by 15.8% at 24 h, mTOR by 19.5% at 24 h, and peroxisome proliferator-activated receptor gamma coactivator 1-alpha (PGC1α) by 12.1% at 24 h. The expressions of 5' AMP-activated protein kinase α (AMPKα, an enzyme that regulates cellular energy homeostasis) was only minimally affected by PTX ($\leq$ 5%) (Fig 13A and 13B).

**Effects of 10 μg/mL PTX on the expression of survival and aging-related proteins.** 10 μg/mL PTX-treated RAW 264.7 cells showed increases in the expression of survival and aging-related proteins, that is, pAKT1/2/3 by 6.5% at 24h, PKC by 8.6% at 24h, pPKC1α by 11.5% at 24h, SP-1 by 8.3% at 48h, a stress responsive deacetylase sirtuin 6 by 5.8% at 24 h, a master regulator of the regulatory pathway in the development and function of regulatory T cells FOXP3 by 7.4% at 12h, a type-I membrane protein related to β-glucuronidase Klotho by 7.1% at 12h, and PLCβ2 by 13.1% at 48h versus the untreated controls. Whereas PTX decreased the expression of telomere reverse transcriptase (TERT) by 10.4% at 12h, SP3 by 16.4% at 48h, sirtuin 1 by 9% at 24h, and FOXP4 by 11% at 48h. The expressions of sirtuin 7, FOXO1 (a transcription factor negatively regulating adipogenesis), and FOXO3 were only minimally affected by PTX ($\leq$ 5%) (Fig 13C and 13D).

**Effects of 10 μg/mL PTX on the expression of endoplasmic reticulum stress proteins.** 10 μg/mL PTX had different effects on the expression of endoplasmic reticulum stress proteins in RAW 264.7 cells. PTX downregulated the proteins contributing to ER stress signaling; eIF2AK3, (PERK, which functions as an ER kinase) by 9.3% at 12 h, eIF2α and p-

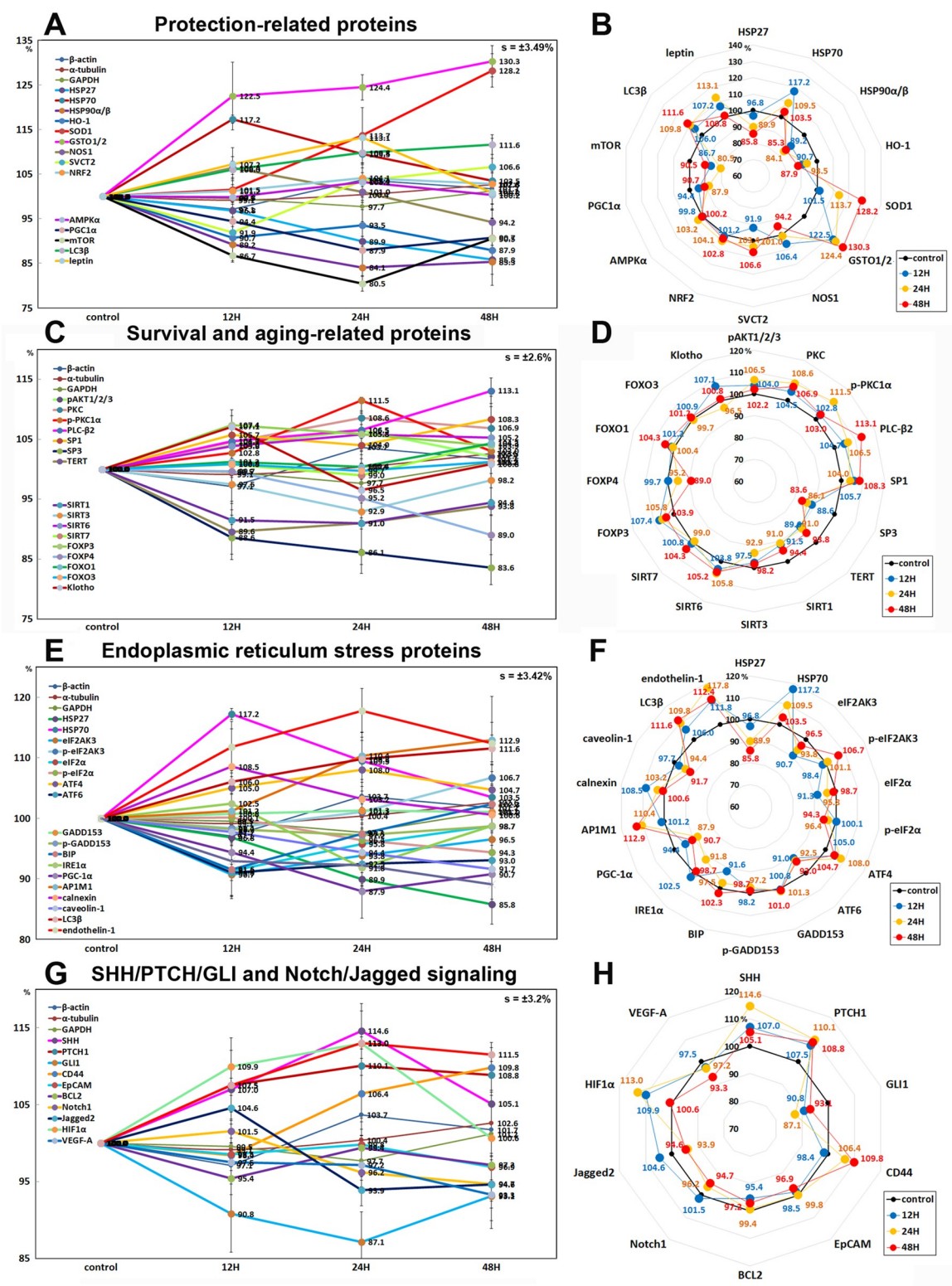

**Fig 13.** Expression of protection-related proteins (A and B), survival and aging-related proteins (C and D), endoplasmic reticulum stress proteins (E and F), and SHH/PTCH/GLI and Notch/Jagged signaling proteins (G and H) in 10 μg/mL PTX-treated RAW 264.7 cells as determined by IP-HPLC. Line graphs, A, C, E, and G show protein expression on the same scale (%) versus culture time (12, 24, or 48 h), whereas the star plots (B, D, F, and H) show the differential expression levels of the proteins at 12, 24, or 48 h after PTX treatment on the appropriate scales (%). The thick black line, untreated controls (100%); the blue, yellow, and red dots show differential protein levels after PTX administration for 12, 24, or 48 h, respectively.

eIF2α (essential factors for protein synthesis also responsible for ER stresses) by 8.7% at 12 h and 5.7% at 48 h, respectively, activating transcription factor 6α (ATF6α) by 9% at 12 h, binding-immunoglobulin protein (BIP, a HSP70 molecular chaperone) by 8.4% at 12 h, serine/threonine-protein kinase/endoribonuclease inositol-requiring enzyme 1 α (IRE1α) by 8.2% at 24 h, PGC-1α by 12.1% at 24 h, caveolin-1 by 8.3% at 48 h, and AIF by 10.9% at 48 h compared to the untreated controls. On the other hand, PTX upregulated the proteins contributing to ER-stress environment in cells; HSP70 by 17.2% at 12 h, p-eIF2AK3 by 6.7% at 48 h, ATF4 (cAMP-response element binding protein 2) by 8% at 24 h, AP1M1 (the medium chain of the trans-Golgi network clathrin-associated protein complex AP-1) by 12.9% at 48 h, calnexin (a chaperone for the protein folding in the membrane of the ER) by 8.5% at 12 h, LC3β (microtubule-associated proteins 1A/1B light chain 3B contributing to autophagosome biogenesis) by 11.6% at 48 h, and endothelin-1 (inducing $Ca^{2+}$ release from ER) by 17.8% at 24 h. The expressions of GADD153 (C/EBP homologous protein (CHOP)) and p-GADD153 were only minimally affected by PTX ($\leq$ 5%) (Fig 13E and 13F).

**Effects of 10 μg/mL PTX on the expression of SHH/PTCH/GLI and Notch/Jagged signaling proteins.** 10 μg/mL PTX was found to influence the expression of SHH/PTCH/GLI signaling proteins positively or negatively in RAW 264.7 cells. PTX upregulated the upstream proteins of SHH/PTCH/GLI signaling; sonic hedgehog (SHH) by 14.6% at 24 h, patched homolog 1 (PTCH1, the receptor for sonic hedgehog) by 8.8% at 48 h, and CD44 (HCAM, the activator of SHH signaling) by 9.8% at 48 h versus the untreated controls, while downregulated the downstream proteins of SHH/PTCH/GLI signaling; GLI1(Glioma-associated oncogene, the effectors of SHH signaling) by 12.9% at 24 h, EpCAM (epithelial cell adhesion molecule, involved in SHH signaling) by 3.1% at 48 h, and BCL2 (GLI binding site in BCL2 promoter, upregulated by GLI1) by 4.6% at 12 h.

On the other hand, PTX downregulated Notch/Jagged signaling proteins; Notch1 (Notch homolog 1, translocation-associated (Drosophila)) by 5.3% at 48 h, Jagged2 (ligand for Notch) by 6.1% at 24 h, and VEGF-A (crosstalk between VEGF and Notch signaling) by 6.7% at 48 h. The expressions of Notch upstream signaling proteins, HIF1α (hypoxia-inducible factors 1α) and Wnt1 were compensatory increased by 13% and 13% at 24 h, respectively (Fig 13G and 13H).

**Effects of 10 μg/mL PTX on the expression of cytodifferentiation proteins.** 10 μg/mL PTX was found to influence the expression of cytodifferentiation proteins positively or negatively in RAW 264.7 cells. PTX upregulated some cytodifferentiation proteins, that is, α-actin by 12% at 24 h, transglutaminase 2 (TGase 2) by 15.3% at 48 h, protein kinase C (PKC, serine/threonine protein kinase, 8.6% at 24 h), p-PKC1α(11.5% at 24 h), focal adhesion kinase (FAK, 6.3% at 48 h), A-kinase anchoring proteins 13 (AKAP13, 7.8% at 48 h), calmodulin (CaM, the universal calcium sensor) by 8.2% at 48 h, calnexin by 8.5% at 12 h, cystatin A (a thiol protease inhibitor) by 6.9% at 12 h, SOX9 (SRY-related HMG-box) by 9.6% at 48 h, AP-1 complex subunit mu-1 (AP1M1, localized at Golgi vesicles for endocytosis and Golgi processing) by 12.9% at 48 h, phosphoinositide-specific phospholipase C β2 (PLCβ2) by 13.1% at 48 h, leucine-rich repeat-containing G-protein coupled receptor 4 (LGR4) by 6.4%, E-cadherin by 17.4% at 48 h, and VE-cadherin (vascular endothelial cadherin) by 12.9%at 12 h, while downregulated other cytodifferentiation proteins, that is, p63 by 9% at 24 h, vimentin by 14.5% at 24 h, TGase 4 by 11% at 24 h, hexokinases 2 (HK2) by 10.9% at 12 h, Krox-25 by 7.1% at 24 h, DLX2 (homeobox protein from distal-less (Dll) gene expressed in the head and limbs of the developing fruit fly) by 17.3% at 24 h, laminin α5 by 14.2% at 48 h, pancreatic lipase by 5.4% at 48 h, and prostate-specific antigen (PSA, gamma-seminoprotein or kallikrein-3 (KLK3)) by 16.9% at 48 h. On the other hand, the expressions of cysteine-rich protein that participates in cytoskeletal remodeling (CRIP1) by 16.3% at 12 h), TBX22 (T-box transcription factor), and

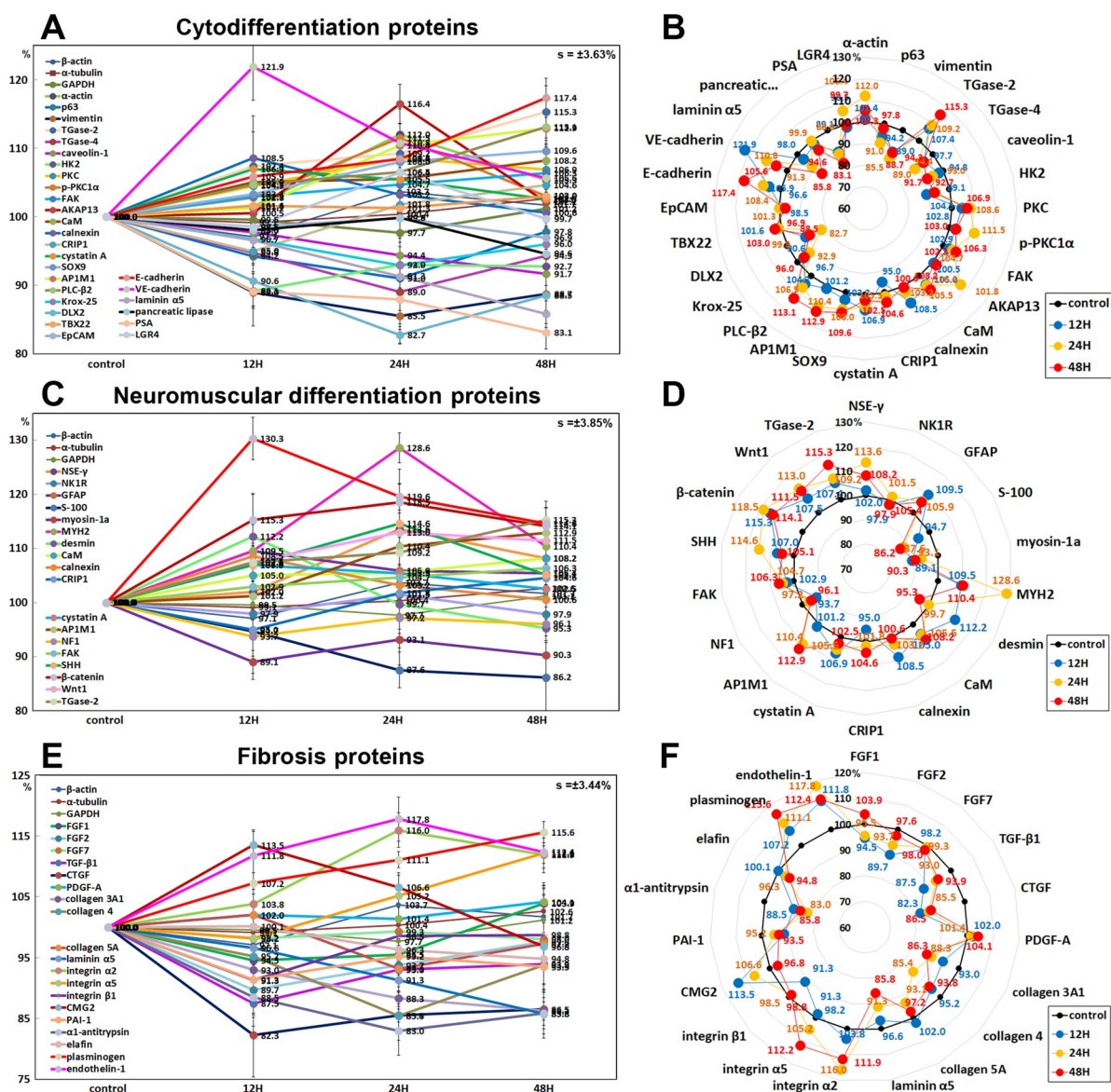

**Fig 14.** Expression of cytodifferentiation proteins (A and B), neuromuscular differentiation proteins (C and D), and fibrosis proteins (E and F) in 10 μg/mL PTX-treated RAW 264.7 cells as determined by IP-HPLC. Line graphs, A, C, and E show protein expression on the same scale (%) versus culture time (12, 24, or 48 h), whereas the star plots (B, D, and F) show the differential expression levels of the proteins at 12, 24, or 48 h after PTX treatment on the appropriate scales (%). The thick black line, untreated controls (100%); the blue, yellow, and red dots show differential protein levels after PTX administration for 12, 24, or 48 h, respectively.

epithelial cell adhesion molecule (EpCAM) were only minimally affected by PTX (≤ 5%) (Fig 14A and 14B).

**Effects of 10 μg/mL PTX on the expression of neuromuscular differentiation proteins.** 10 μg/mL PTX was found to have positive influence on the expression of neuromuscular differentiation proteins in RAW 264.7 cells. PTX upregulated some neuromuscular differentiation proteins, that is, neuron specific γ enolase (NSEγ) by 13.6% at 24 h, glial fibrillary acidic protein (GFAP) by 9.5% at 12h, myosin heavy chain 2 (MYH2) by 28.6% at 24 h, desmin by 12.2% at 12h, calmodulin (CaM) by 8.2% at 48 h, calnexin by 8.5% at 12 h, cystatin A by 6.9% at 12 h, AP-1 complex subunit mu-1 (AP1M1) by 12.9% at 48 h, focal adhesion

kinase (FAK, substrate for tyrosine kinase of Src) by 6.3% at 48 h, SHH by 14.6 at 24 h, β-catenin by 18.5% at 24 h, Wnt1 by 13% at 24 h, TGase 2 by 15.3% at 48 h, and α-smooth muscle actin (α-SMA) by 30.3% at 12 h, while downregulated other neuromuscular differentiation proteins, that is, S-100 by 13.8% at 48 h, unconventional myosin-1a (membrane binding class I myosin), 10.9% at 12 h, neurofibromin 1 (NF1) by 6.3% at 12 h. The expressions of neurokinin 1 receptor (NK1R, substance P receptor) and cysteine-rich protein 1 (CRIP1) were only minimally affected by PTX ($\leq$ 5%) (Fig 14C and 14D).

**Effects of 10 µg/mL PTX on the expression of fibrosis proteins.** 10 µg/mL PTX was found to decrease the expression of fibrosis-inducing proteins; FGF1 by 5.5% at 12 h, FGF2 by 10.3% at 12 h, TGF-β1 by 12.5% at 12 h, CTGF by 17.7% at 12 h, collagen 3A1 by 13.7% at 48 h, collagen 4 by 14.6% at 24 h, collagen 5A by 6.9% at 24 h, laminin α5 by 14.2% at 48 h, integrin β1 by 8.7% at 12 h, plasminogen activator inhibitor-1 (PAI1) by 8.7% at 12 h, α1-antitrypsin by 7% at 24 h, elafin (peptidase inhibitor 3, elastase-specific protease inhibitor) by 5.2% at 48 h, and also to increase anti-fibrosis proteins; plasminogen by 15.6% at 48 h, integrin α2 by 16% at 24 h, integrin α5 by 12.2% at 48 h, and capillary morphogenesis gene 2 (CMG2) by 13.5% at 12 h. On the other hand, the expression of endothelin-1, a key role of vascular homeostasis, was reactively upregulated by 17.8% at 24 h. And the expressions of FGF7 and platelet-derived growth factor A (PDGF-A) were only minimally affected by PTX ($\leq$ 5%) (Fig 14E and 14F).

**Effects of 10 µg/mL PTX on the expression of oncogenesis proteins.** 10 µg/mL PTX was found to influence the expression of oncogenesis proteins positively or negatively in RAW 264.7 cells. PTX decreased the expression of tumor suppressor proteins; breast cancer type 1 susceptibility protein (BRCA1) by 8.4% at 12 h, breast cancer type 2 susceptibility protein (BRCA2) by 18.6% at 24 h, neurofibromin 1 (NF1, a GTPase-activating protein that negatively regulates RAS/MAPK pathway activity) by 6.3% at 12 h, ataxia telangiectasia caused by mutations (ATM, a serine/threonine protein kinase recruited and activated by DNA double-strand breaks) by 13.9% at 24 h, maspin (a mammary serine protease inhibitor, serpin superfamily) by 9.5% at 24 h, deleted in malignant brain tumors 1 protein (DMBT1, a glycoprotein that interacts between tumor cells and the immune system) by 18% at 24 h, methyl-CpG-binding domain protein 4 (MBD4, a DNA repair enzyme that removes mismatched U or T) by 6.7% at 12 h, p53 by 17.5% at 24 h, retinoblastoma protein (Rb1) by 9.8% at 48 h, but increased the expression of PTCH1 (Protein patched homolog 1, a suppressor of smoothened release, which signals cell proliferation) by 10.1% at 24 h. On the other hand, PTX increased the expression of oncogenic proteins; carcinoembryonic antigen (CEA) by 9.6% at 12 h, 14-3-3 θ proteins (a phosphoserine binding protein that regulates Cdc25C) by 10.9% at 48 h, survivin (a negative regulator of apoptosis) by 11.9% at 48 h, mucin 4 (an anti-adhesive glycoprotein that contributes to tumor development and metastasis) by 15.6% at 24 h, Yes-associated protein 1 (YAP1, a potent oncogene that binds to 14-3-3) by 10.3% at 12 h, but decreased the expression of mucin 1 (a glycoprotein with extensive O-linked glycosylation of its extracellular domain, oncogenic epithelial membrane antigen) by 21.2% at 48 h. the expression of phosphatase and tensin homolog (PTEN, tumor suppressor protein) and PIM1 (proto-oncogene serine/threonine-protein kinase) were only minimally affected by PTX ($\leq$ 5%) (Fig 15A and 15B).

**Effects of 10 µg/mL PTX on the expression of angiogenesis proteins.** 10 µg/mL PTX reduced the expression of major angiogenesis proteins in RAW 264.7 cells; angiogenin by 5.3% at 12 h, vascular endothelial growth factor A (VEGF-A) by 6.7% at 48 h), VEGF-D by 15.6% at 24 h, von Willebrand factor (vWF) by 7.7% at 12 h, Fms-related tyrosine kinase 4 (FLT4) by 12.4% at 48 h, lymphatic vessel endothelial hyaluronan receptor 1 (LYVE1) by 10.9% at 48 h, fibroblast growth factor-2 (FGF2) by 10.3% at 12 h, CD106 (vascular cell adhesion molecule-1 (VCAM-1) by 15.5% at 12 h, matrix metalloprotease-2 (MMP2), 14.9% at 24

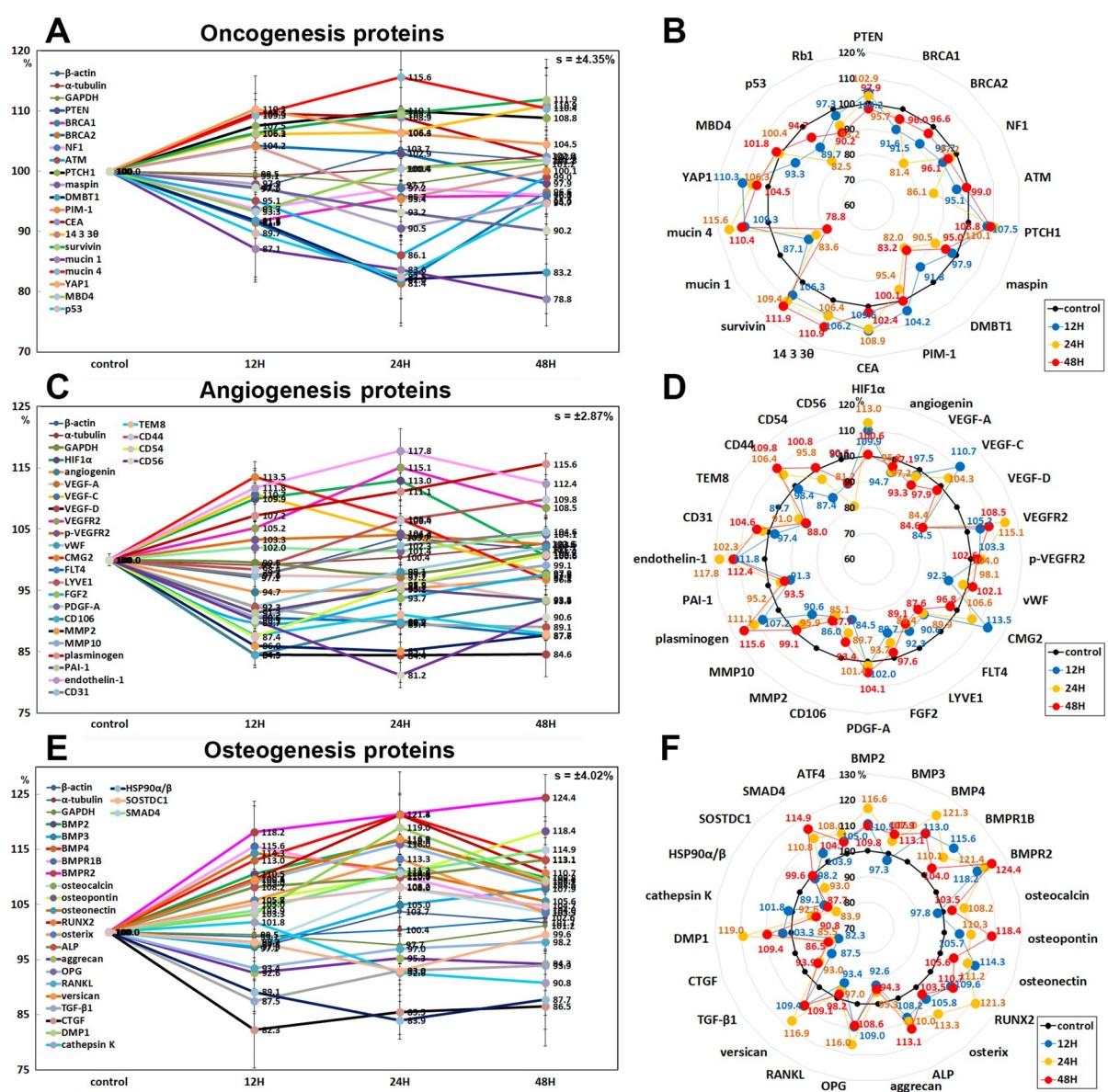

**Fig 15.** Expression of oncogenesis proteins (A and B), angiogenesis proteins (C and D), and osteogenesis proteins (E and F) in 10 μg/mL PTX-treated RAW 264.7 cells as determined by IP-HPLC. Line graphs, A, C, and E show protein expression on the same scale (%) versus culture time (12, 24, or 48 h), whereas the star plots (B, D, and F) show the differential expression levels of the proteins at 12, 24, or 48 h after PTX treatment on the appropriate scales (%). The thick black line, untreated controls (100%); the blue, yellow, and red dots show differential protein levels after PTX administration for 12, 24, or 48 h, respectively.

h, MMP-10 (9.4% at 12 h), plasminogen activator inhibitor-1 (PAI1) by 8.7% at 12 h, tumor-specific endothelial marker 8 (TEM8) by 12.3% at 12 h, CD54 (intercellular adhesion molecule 1 (ICAM-1)) by 12.6% at 12 h, and CD56 (neural cell adhesion molecule (NCAM)) versus the untreated controls, while it compensatory elevated the expression of wound healing-related proteins, VEGF-C [42] by 10.7% at 12 h, VEGFR2 [43] by 15.1% at 24 h, capillary morphogenesis protein 2 (CMG2) [44] by 13.5% at 12 h, plasminogen (cleavage of the serine proteinase plasminogen to form plasmin, proangiogenic proteinase) by 15.6% at 48 h, endothelin-1 (21-amino acid vasoconstricting peptides) [45] by 17.8% at 24 h, and CD44 (homing cell

adhesion molecule (HCAM)) by 9.8% at 24 h. The expression of PDGF-A and CD31 (platelet endothelial cell adhesion molecule (PECAM-1)) were only minimally affected by PTX ($\leq$ 5%) (Fig 15C and 15D).

**Effects of 10 µg/mL PTX on the expression of osteogenesis proteins.**   10 µg/mL PTX-treated RAW 264.7 cells showed increase in the expression of osteogenesis proteins, that is, bone morphogenetic protein-2 (BMP2, 16.6% at 24 h), BMP3 (a negative regulator for bone density by antagonizing other BMPs) by 7.9% at 24 h, BMP4 by 21.3% at 24 h, BMPR1B (bone morphogenetic protein receptor type-1B) by 15.6% at 12 h, BMPR2 (bone morphogenetic protein receptor type 2) by 24.4% at 48 h, osteocalcin by 8.2% at 24 h, osteopontin by 18.4% at 48 h, osteonectin by 14.3% at 12 h, mammalian Runt-related transcription factor 2 (RUNX2, a key transcription factor associated with osteoblast differentiation) by 21.3% at 24 h, osterix (a zinc finger-containing transcriptional activator for osteoblastic differentiation) by 13.3% at 48 h, alkaline phosphatase (ALP) by 13.1% at 48 h, osteoprotegerin (OPG) by 16% at 24 h, versican (abundant in the woven bone matrix) by 16.9% at 24 h, DMP1 by 19% at 24 h, SMAD4 by 14.9% AT 48 H, and activating transcription factor 4 (ATF4) by 8% at 24 h, while decreased the expression of aggrecan (a large chondroitin sulfate proteoglycan) by 7.4% at 12 h, receptor activator of nuclear factor kappa-B ligand (RANKL, a binding partner of OPG) by 6.6% at 12 h, TGF-β1 by 12.5% at 12 h, connective tissue growth factor (CTGF, CCN2, a role in chondrogenesis and angiogenesis) by 17.7% at 12 h, cathepsin K (a lysosomal cysteine protease involved in bone remodeling and resorption [46]) by 9.2% at 48 h, HSP90α/β (a crucial regulator of vesicular transport of cellular proteins in osteoclasts [47]) by 16.1% at 24 h, and sclerostin domain-containing protein 1 (SOSTDC1, a bone morphogenetic protein antagonist) by 7% at 24 h (Fig 15E and 15F).

## Global protein expressions in 10 µg/mL PTX-treated RAW 264.7 cells

Fig 16 presents the global protein expression changes in 150 representative proteins of 21 different protein signaling pathways as a star plot. 10 µg/mL PTX was found to affect the expression of proteins in different signaling pathways of RAW 264.7 cells, and regulated characteristic cellular functions. PTX increased cell proliferation by upregulating proliferation-activating proteins, Ki-67, PCNA, cyclin D2, and CDK4, and enhancing proliferation-related pathways; that is, cMyc/MAX/MAD network by upregulating cMyc and MAX, p53/Rb/E2F by upregulating E2F1 and downregulating p-Rb1, and Wnt/β-catenin signaling by upregulating Wnt1, APC, β-catenin, TCF1 and by downregulating snail and AXIN2. PTX decreased histone/DNA methylation by downregulating EZH2, DNMT1, and DMAP1 and upregulating KDM4D, increased histone acetylation by upregulating PCAF and HMGB1, and downregulating HDAC-10, and also enhanced protein translation by upregulating DOHH, DHS, and eIF5A1.

PTX was found to downregulate TGF-β1, TGF-β2, TGF-β3, SMAD2/3, HGFα, Met, but reactively upregulate SMAD4, while it stimulated RAS signaling by upregulating KRAS, HRAS, pAKT1/2/3, PI3K, Rab1, ERK1, and pERK1 and downregulating JAK2, p-JNK1, and MEKK1, and simultaneously elevated NFkB signaling by upregulating NFkB, GADD45, p38, p-p38, and MDR, and downregulating p-mTOR and PGC1α.

PTX increased the expression of acute inflammatory proteins (TNFα, IL-1, IL-6, granzyme B, MCP1, CXCR4, CTLA4, M-CSF, and MMP1), cell-mediated immunity proteins (CD4, CD8, CD80, HLA-DR, and perforin), and innate immunity proteins (lactoferrin, versican [48], TLR3, and TLR4).

PTX diminished p53-mediated apoptosis by downregulating p53, BAD, NOXA, APAF1, caspase 9, c-caspase 9, and c-PARP in RAW 264.7 cells, whereas it enhanced FAS-mediated

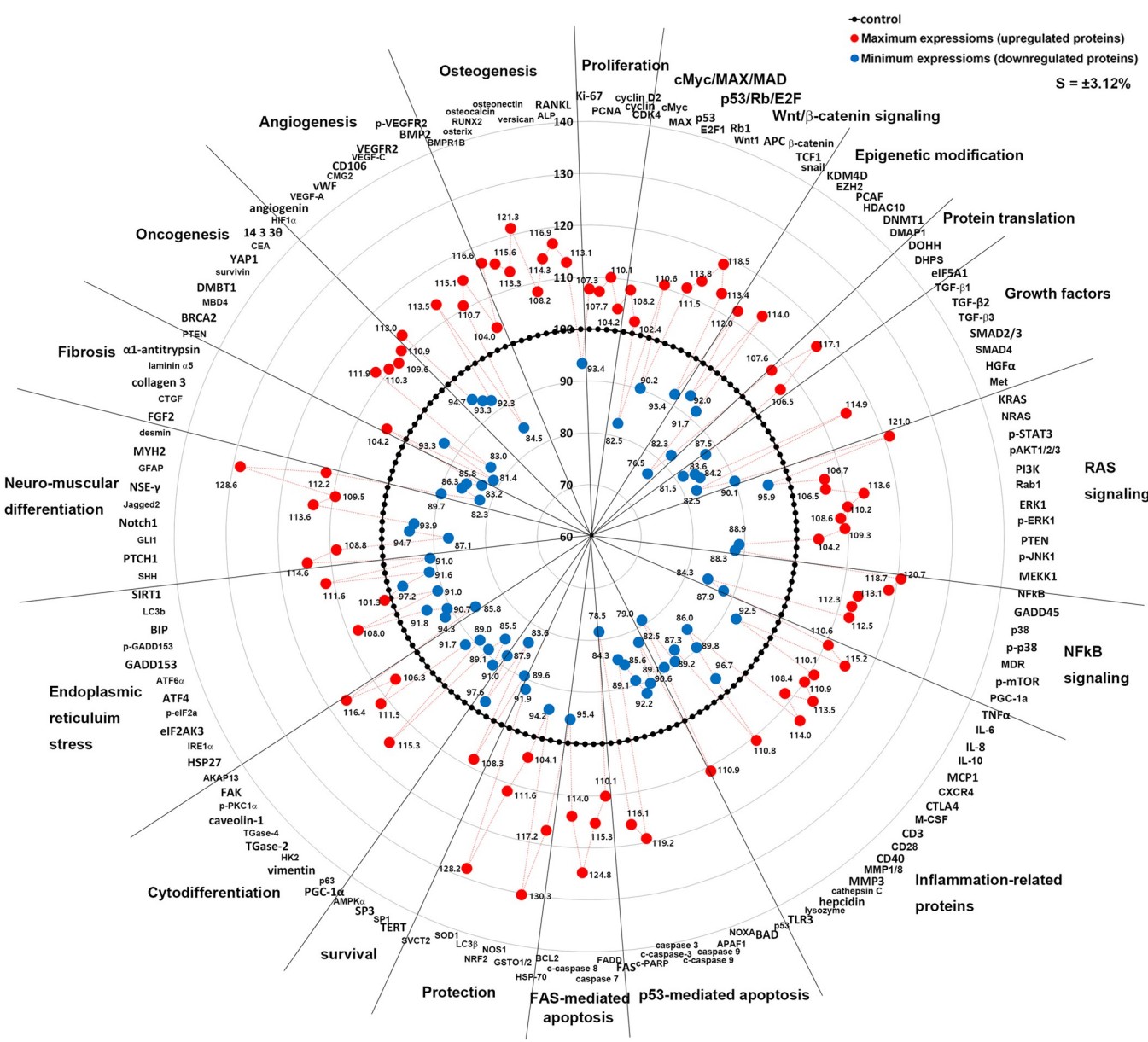

**Fig 16. Star plot of global protein expression in RAW 264.7 cells treated with 10 μg/mL PTX.** The representative proteins (n = 150) were selected and their maximum or minimum expression levels (%) were plotted in a circular manner. 21 major signaling pathways showed different levels of protein expression. 10 μg/mL PTX activated the growth factors, epigenetic modification, protein translation, RAS and NFkB signaling, protection, neuromuscular ad osteoblastic differentiation, acute inflammation associated with innate immunity and cell-mediated immunity, but inactivated ER stress, fibrosis, and chronic inflammation. FAS-mediated apoptosis was enhanced contrary to p53-mediated apoptosis. Also noted that the upregulation of oncogenic proteins and the downregulation of tumor suppressor proteins. Red circle: maximum expression of upregulated proteins. Blue circle: minimum expression of downregulated proteins.

apoptosis by upregulating FAS, FADD, caspase 7, c-caspase 8, c-caspase 10, caspase 3, and c-caspase 3. On the other hand, PTX activated cell protection by increasing the expression of HSP70, SOD1, GSTO1/2, SVCT2, NRF2, and NOS1 [49], and stimulated cell survival and homeostasis by upregulating PLCβ2, SP1, and downregulating TERT, SIRT1, SP3, AMPKα, and PGC1α. Particularly, PTX attenuated ER-stress by downregulating HSP27, IRE1α, eIF2AK3, p-eIF2α, ATF6α, p-GADD153, caveolin-1, AIF, BIP, and upregulating LC3β.

PTX stimulated SHH/PTCH signaling by upregulating SHH and PTCH1, but reduced the expression of GLI1, and subsequently attenuated Notch/Jagged signaling by downregulating Notch1 and Jagged2. PTX-treated RAW 264.7 cells appeared to be differentiated into mature macrophages by upregulation of α-actin, VE-cadherin, CaM, TGase 2, PKC, p-PKC1α, AP1M1, FAK, AND AKAP13, and showed a characteristics of neuromuscular differentiation by upregulating NSE-γ, GFAP, MYH2, and desmin.

PTX was found to have anti-fibrosis effect on RAW 264.7 cells by downregulating FGF2, CTGF, collagen 3A1, laminin α5, and α1-antitrypsin. And PTX-treated cells showed a state of oncogenic stress by upregulating survivin, YAP1, CEA, and 14-3-3θ, and downregulating ATM, BRCA2, MBD4, and DMBT1.

PTX affected the expression of angiogenesis proteins in RAW 264.7 cells positively or negatively, that is, angiogenin, VEGF-A, vWF, and CD106 (VCAM-1) were downregulated by 10 μg/mL PTX, while HIF1α, CMG2, VEGF-C, VEGFR2, and p-VEGFR2 were upregulated. On the other hand, PTX consistently increased the expression of osteogenesis proteins, BMP2, BMPR1B, osterix, RUNX2, osteocalcin, osteonectin, versican, and ALP, but decreased RANKL expression (Fig 16).

## Comparison between 10 μg/mL and 300 μg/mL PTX application in RWA 264.7 cells

**Proliferation-related protein expression by 10 μg/mL or 300 μg/mL PTX.** 300 μg/mL PTX reduced the expression of proliferation-related proteins in RAW 264.7 cells compared to 10 μg/mL PTX, that is, Ki-67 by 8.5% at 48 h, CDK4 by 9.9% at 12 h, cyclin D2 by 16.6% at 12 h, E2F1 by 14.8% at 24 h, cMyc by 6.4% at 12 h, MAX by 5.1% at 12 h, and MAD1 by 8.6% at 48 h, but increased the expression of p-Rb1 by 2.7% at 24 h. 10 μg/mL PTX stimulated the proliferation of RAW 264.7 cells by upregulating Ki-67, CDK4, cyclin D2, E2F1, cMyc, MAX, and downregulating p-Rb1, while 10 μg/mL PTX showed anti-proliferative effect on RAW 264.7 cells by downregulating Ki-67, cyclin D2, E2F1, and MAX compared to the untreated controls (Fig 17A and 17B).

**RAS/NFkB signaling protein expression by 10 μg/mL or 300 μg/mL PTX.** 300 μg/mL PTX reduced the expression of RAS/NFkB signaling proteins in RAW 264.7 cells compared to 10 μg/mL PTX, that is, KRAS by 11.3% at 12 h, pAKT1/2/3 by 11.3% at 12 h, p-ERK1 by 14.7% at 24 h, NFkB by 15.4% at 24 h, p-p38 by 6.9% at 24 h, GADD153 by 10.8% at 48 h, and p-PKC1α by 5.8% at 24 h. However, the expression of pAKT1/2/3 and p-PKC1α by 300 μg/mL PTX were increased to 102% at 48h and 106.9% at 12 h, respectively, compared to those by 10 μg/mL PTX. 10 μg/mL PTX enhanced RAS/NFkB signaling by upregulating KRAS, pAKT1/2/3, p-ERK1, NFkB, p-p38, GADD153, and p-PKC1α compared to the untreated controls, while 300 μg/mL PTX attenuated RAS/NFkB signaling by downregulating pAKT1/2/3, p-ERK1, NFkB, p-p38, and GADD153 (Fig 17C and 17D).

**Inflammatory protein expression by 10 μg/mL or 300 μg/mL PTX.** 300 μg/mL PTX markedly decreased the expression of inflammation-associated proteins in RAW 264.7 cells compared to 10 μg/mL PTX, that is, TNFα by 11% at 12 h, CD4 by 21.8% at 24 h, CD80 by 6.9% at 24 h, M-CSF by 15% at 48 h, CXCR4 by 7.1% at 24 h, MMP1 by 19.3% at 48 h, and lactoferrin by 15.7% at 24 h, but increased the expression of an inflammation suppressor TGF-β1 by 23.5% at 48 h. It was evident 10 μg/mL PTX stimulated the inflammatory reaction of RAW 264.7 cells by upregulating TNFα, IL-6, CD4, CD80, M-CSF, CXCR4,MMP1, lactoferrin, and downregulating TGF-β1 compared to the untreated controls, while 300 μg/mL PTX showed anti-inflammatory effect on cells by downregulating TNFα, IL-6, CD4, M-CSF, MMP1, lactoferrin, and upregulating TGF-β1 (Fig 17E and 17F).

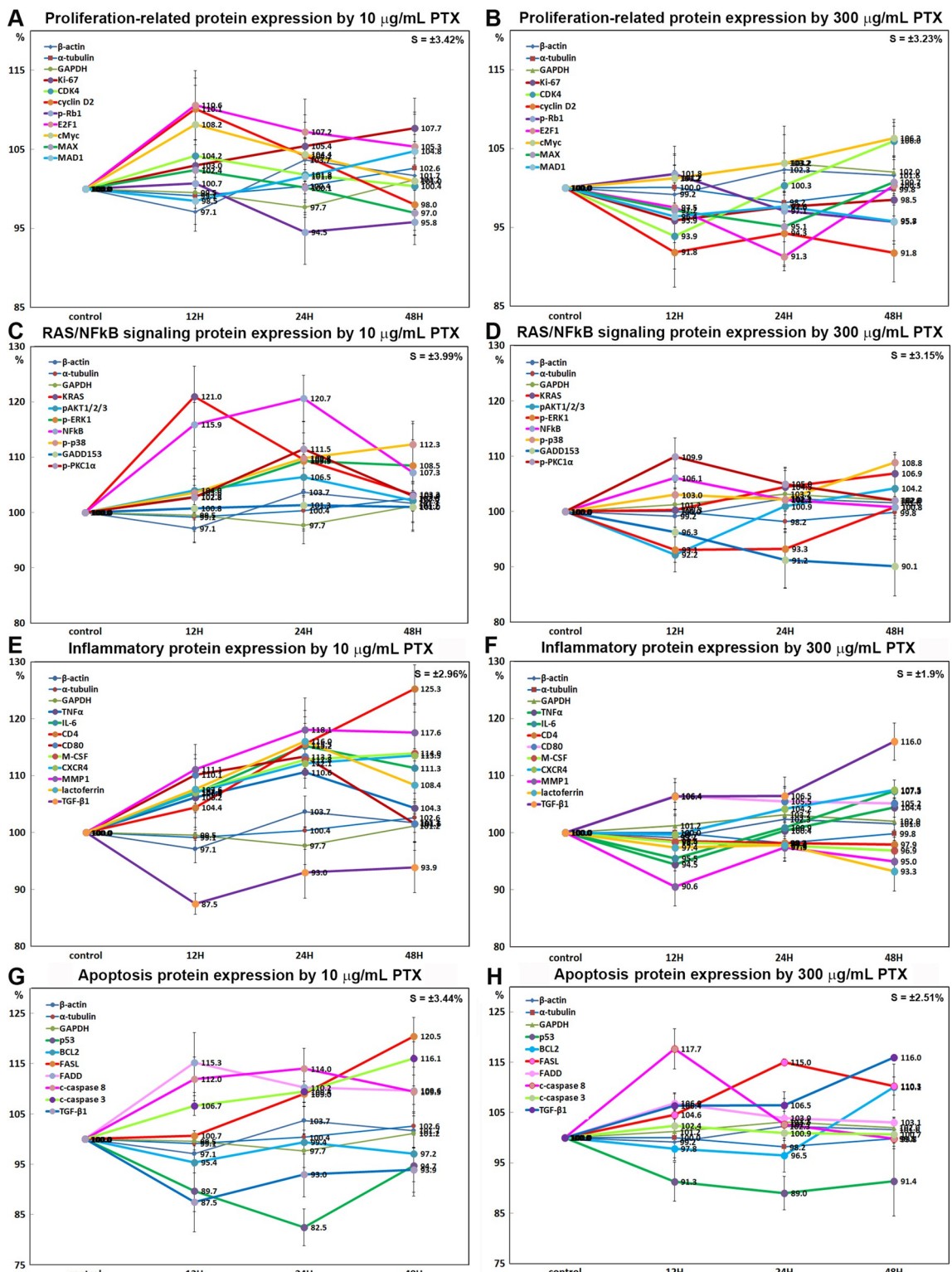

**Fig 17. IP-HPLC comparison between the protein expression of 10 μg/mL and 300 μg/mL PTX-treated RAW 264.7 cells.**
Expression of proliferation-related proteins (A and B), RAS/NFkB signaling proteins (C and D), inflammatory proteins (E and F), and apoptosis proteins (G and H) as determined by IP-HPLC. Line graphs, A, C, E, and G show 10 μg/mL PTX-induced protein expressions, whereas B, D, F, H show 300 μg/mL PTX-induced protein expressions on the same scale (%) versus culture time (12, 24, or 48 h).

**Apoptosis protein expression by 10 μg/mL or 300 μg/mL PTX.** 300 μg/mL PTX was found to have different effect on the expression of oncogenesis proteins in RAW 264.7 cells from 10 μg/mL PTX. The expressions of tumor suppressor protein p53, anti-apoptotic protein BCL2, apoptosis trigger FASL, and TGF-β1 by 300 μg/mL PTX were higher by 7.9% at 24 h, 13.3% at 48 h, 5.5% at 24 h, and 23.5% at 48 h, respectively, than those by 10 μg/mL PTX, while the expressions of FAS-associated protein with death domain (FADD), apoptosis executioners c-caspase 8 and c-caspase 3 by 300 μg/mL PTX were lower by 7.3% at 12 h, 9.9% at 24 h, and 13.3% at 48 h, respectively, than those by 10 μg/mL PTX. 10 μg/mL PTX induced significant apoptosis effect on RAW 264.7 cells through FAS-mediated apoptosis by upregulating FAS, FADD, c-caspase 8, and c-caspase 3 compared to the untreated controls, while 300 μg/mL PTX slightly reduced the apoptosis effect by downregulating FADD, c-caspase 8, and c-caspase 3 compared to 10 μg/mL PTX (Fig 17G and 17H).

**ER stress protein expression by 10 μg/mL or 300 μg/mL PTX.** 300 μg/mL PTX increased the expression of ER stress proteins in RAW 264.7 cells compared to 10 μg/mL PTX, that is, eIF2AK3 (PERK) by 14.9% at 12 h, p-eIF2α by 12.8% at 48 h, ATF6α by 13.7% at 48 h, BIP by 10.3% at 12 h. and IRE1α by 3.3% at 24 h. And 300 μg/mL PTX minimally affected the expression of GADD153 (CHOP) compared to 10 μg/mL PTX. It was found that 10 μg/mL PTX reduced ER stress by downregulating eIF2AK3, p-eIF2α, ATF6α, BIP, and IRE1α, while 100 μg/mL PTX alleviated the reduction of ER stress by upregulating eIF2AK3, p-eIF2α, ATF6α, and BIP compared to 10 μg/mL PTX, particularly, the expression of eIF2AK3, p-eIF2α, ATF6α, and BIP were increased by 300 μg/mL PTX to 106% at 48 h, 106.4% at 48 h, 105.8% at 48 h, an 104.4% at 24 h, respectively, versus the untreated controls (Fig 18A and 18B).

**Fibrosis protein expression by 10 μg/mL or 300 μg/mL PTX.** 300 μg/mL PTX increased the expression of fibrosis proteins in RAW 264.7 cells compared to 10 μg/mL PTX, that is, FGF1 by 7.1% at 24 h, FGF2 by 8.3% at 12 h, CTGF by 15.6% at 12 h, collagen 3A1 by 29.1% at 24 h, laminin α5 by 28% at 48 h, and α1-antitrypsin by 21.6% at 24 h. 10 μg/mL PTX showed potent anti-fibrosis effect on RAW 264.7 cells by downregulating FGF1, FGF2, CTGF, collagen 3A1 and 5A, laminin α5, and α1-antitrypsin, while 300 μg/mL PTX showed no anti-fibrosis effect rather increased the expression of FGF1 (by 6.4% at 48 h), collagen 3A1 (14% at 24 h), collage 5A (4.6% at 24 h), and laminin α5 (9.8% at 48 h) compared to the untreated controls (Fig 18C and 18D).

**Neuromuscular differentiation protein expression by 10 μg/mL or 300 μg/mL PTX.** 300 μg/mL PTX decreased the expression of neuromuscular differentiation proteins in RAW 264.7 cells compared to 10 μg/mL PTX, that is, NSEγ by 7.6% at 48 h, GFAP by 5.2% at 12 h, MYH2 by 18.1% at 24 h, desmin by 8.5% at 24 h, and α-SMA by 21.5% at 12 h, but the expression of myosin 1a (membrane binding class I myosin) was increased by 11.3% at 12 h. 10 μg/mL PTX induced nerve differentiation by upregulating NSEγ and GFAP, and muscle differentiation by upregulating MYH2 (skeletal muscle heavy chain 2), desmin, and α-SMA, while 300 μg/mL PTX only slightly increased the expression of NSEγ, GFAP, MYH2, desmin and α-SMA compared to the untreated controls (Fig 18E and 18F).

**Osteogenesis protein expression by 10 μg/mL or 300 μg/mL PTX.** 300 μg/mL PTX decreased the expression of osteogenesis proteins in RAW 264.7 cells compared to 10 μg/mL PTX, that is, BMP2 by 25.8% at 48 h, RUNX2 by 27.7% at 12 h, osterix by 8.7% at 24 h, osteocalcin by 5.2% at 24 h, osteopontin by 11.3% at 48 h, and osteonectin by 9.4% at 24 h. However, the expression ratios of OPG and RANKL, which are essential signaling molecules of RANKL/RANK/OPG system regulating osteoclast differentiation/activation and calcium release from the skeleton, were increased by 6.6% and 9.6% at 12 h, respectively. It was found 10 μg/mL PTX showed strong osteogenic effect on RAW 264.7 cells by upregulating BMP2, RUNX2, osterix, osteocalcin, osteopontin, osteonectin, and OPG, while 300 μg/mL PTX alleviated the

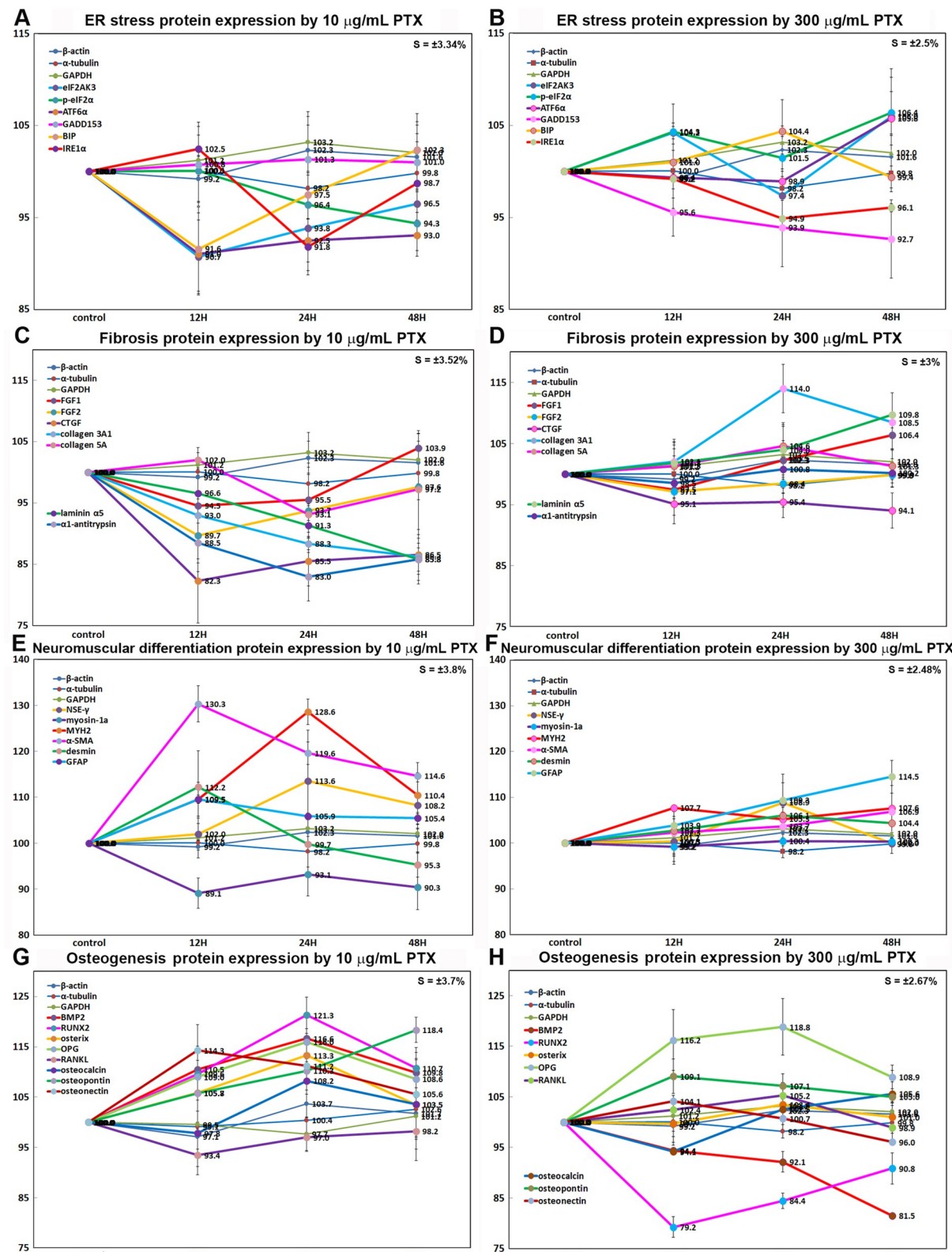

**Fig 18. IP-HPLC comparison between the protein expression of 10 μg/mL and 300 μg/mL PTX-treated RAW 264.7 cells.** Expression of ER stress proteins (A and B) fibrosis proteins (C and D), neuromuscular differentiation proteins (E and F) and osteogenesis proteins (G and H) as determined by IP-HPLC. Line graphs, A, C, E, and G show 10 μg/mL PTX-induced protein expressions, whereas B, D, F, H show 300 μg/mL PTX-induced protein expressions on the same scale (%) versus culture time (12, 24, or 48 h).

osteogenic effect by downregulating RUNX2, BMP2, and osteocalcin compared to the untreated controls (Fig 18G and 18H).

## Discussion

The pharmacological effect of PTX was frequently investigated in different cell types including RAW 264.7 cells [19–22] and animals [50–53] by using higher dose PTX, 100–500 μg/mL, rather than the therapeutic dose in human (about 10 μg/mL). It was also reported that the low dose PTX, 10, 25, 50 mg/kg, led to an increase in the expression of caspase 3 and TNFα in the rat hippocampus following lipopolysaccharide (LPS)-induced inflammation [54]. And the TNFα production by lipopolysaccharide (LPS)-stimulated human alveolar macrophages was significantly suppressed in the presence of PTX at concentration of 2 mM and 1 mM (278.3 μg/mL), but not at 0.5 mM, 0.1mM (27.8 μg/mL), and 0.01 mM, while production of IL-1β, IL-6, and GM-CSF remained unaffected. These data indicate PTX showed anti-inflammatory effect selectively depending on its concentration [55].

In the present study, RAW 264.7 cells, which are originally murine monocytes, were explored for PTX-induced protein expression changes by administrating with two different doses, 10 μg/mL PTX similar to the therapeutic dose in human, and 300 μg/mL PTX which was frequently used in cell and animal experiments. First of all, the 10 μg/mL PTX-induced effect was compared with the 300 μg/mL PTX-induced effect. However, in clinical application, even the therapeutic dose PTX, about 10 μg/mL, produces diverse side effects including belching, bloating, stomach discomfort or upset, nausea, vomiting, indigestion, dizziness, flushing, angina, palpitations, hypersensitivity, itchiness, rash, hives, bleeding, hallucinations, arrhythmias, and aseptic meningitis [56,57]. Therefore, it is thought that the low dose 10 μg/mL PTX-induced protein expression may be more informative to know the real pharmacological effect of PTX in human than the high dose 300 μg/mL PTX-induced protein expression in cell culture. Therefore, in the present study, 10 μg/mL PTX-induced effect on cells was more extensively investigated than 300 μg/mL PTX-induced effect.

In the global protein expression of RAW 264.7 cells by 10 μg/mL PTX, a competitive non-selective phosphodiesterase inhibitor which is known to raise intracellular cAMP and activate PKA, actually enhanced RAS signaling by upregulating AKAP13, pAKT1/2/3, PKC, p-PKC1α, KRAS, and HRAS, and subsequently activated histone/DNA demethylation and acetylation by upregulating KDMD4, HMGB1, and PCAF, and downregulating HDAC10, MBD4, DMAP1, DNMT1 and EZH2, and subsequently induced cellular proliferation by upregulating cMyc/MAX/MAD network proteins, Wnt/β-catenin signaling objective protein TCF1, proliferation activating proteins Ki-67, PCNA, PLK4, cyclin D2, and cdc25A in this study. On the other hand, the expression of CDK inhibitors, p14, p15/16, and p21 were compensatory upregulated. The double IP-HPLC to assess the amount of protein complex containing two different target proteins showed the increase of cMyc-MAX heterodimer (Fig 8E and 8F) and β-catenin-TCF1 complex (Fig 9G and 9H) which led to cell cycle progression [58], and the decrease of cMyc-MAD1 heterodimer and CDK4-p27 complex concomitantly. On the other hand, the double IP-HPLC also revealed the increase of E2F1-Rb1 and CDK4-p21 complexes (Fig 9C and 9D), which mitigated cell cycle progression [59].

Although the expression of cMyc and MAX were decreased by 10 μg/mL PTX at 12, 24, and 48 h, the expression of MAD1 was increased in western blot and IP-HPLC, and the double IP-HPLC showed dominant increase of cMyc-MAX heterodimer and decrease of cMyc-MAD heterodimer versus the untreated controls (Fig 8E and 8F). Therefore, it is suggested that cMyc and MAX are competitively utilized to form cMyc-MAX heterodimer against cMyc-MAD heterodimer at 12, 24, and 48 h after 10 μg/mL PTX treatment, and then the unbound

cMyc and MAX are gradually reduced in amount as detected in western blot (Fig 5) and IP-HPLC. And more, although the expression of E2F1 was increased by 10 μg/mL PTX at 12, 24, and 48 h, the phosphorylated Rb1 (p-Rb1) was decreased, and both E2F1-Rb1 and CDK4-p21 complexes were increased in double IP-HPLC. Therefore, it is suggested p53/Rb/E2F signaling rarely exerts to enhance cell proliferation in 10 μg/mL PTX-treated RAW 264.7 cells.

Cell counting assay of 10 μg/mL PTX-treated cells showed the increase of cell number at 12, 24, and 48 h. Regarding proliferation-related proteins, ICC revealed strong positive reaction of Ki-67 at 12, 24, and 48 h compared to the untreated controls, and western blot showed strong bands of Ki-67, E2F1, Wnt1, and TCF1 at 12, 24, and 48 h. Taken together, it is evident that 10 μg/mL PTX enhances RAS signaling, and subsequently activates histone/DNA demethylation and acetylation, cMyc/MAX/MAD network, and Wnt/β-catenin signaling, and resulted in the proliferation of RAW 264.7 cells. Whereas 300 μg/mL PTX showed a trend to decrease the expression of proliferation-related proteins, Ki-67, cMyx, MAX, p-Rb1, E2F1, and cyclin D2 compared to 10 μg/mL PTX.

The non-selective phosphodiesterase inhibitor, PTX specifically induced RAS signaling, and subsequently stimulated NFkB signal by upregulating PKC, p-PKC-1α, AKAP13, MDR, GADD45, p38, and p-p38, and eventually influenced on the expression of protection-, ER stress-, apoptosis-, and inflammatory proteins in RAW 264.7 cells. Regarding cellular protection, 10 μg/mL PTX upregulated antioxidant proteins, SOD1, GSTO1/2, and SVCT2, but downregulated NRF2 regulating antioxidant expression, PGC-1α regulating mitochondrial biogenesis, and AMPK1α that plays a role in cellular energy homeostasis. And 10 μg/mL PTX mitigated ER stress by downregulating eIF2AK3, p-eIF2AK3, eIF2α, p-eIF2α, GADD153, p-GADD153, BIP, IRE1α, and ATF6α. Therefore, it is suggested that 10 μg/mL PTX-treated in RAW 264.7 cells are relatively safe under control of cellular protection and homeostasis. And more, the 10 μg/mL PTX-induced RAS signal by cAMP accumulation activated epigenetic modification through histone/DNA demethylation or acetylation, and subsequently elevated protein translation, which eventually positively influenced on cell proliferation, differentiation, protection and survival of RAW 264.7 cells.

10 μg/mL PTX-induced NFkB signal also influenced on the expression of inflammatory protein positively or negatively in RAW 264.7 cells. It was found some acute inflammatory proteins (TNFα, IL-1, IL-6, MCP1, CXCR4, granzyme B, and MMP1), innate immunity proteins (β-defensin 1, lactoferrin, versican, TLR 3 and 4), and cell-mediated immunity proteins (CD4, CD8, CD80, HLA-DR, perforin, and CTLA4) were upregulated, while some chronic inflammatory proteins including IL-12, CD68, CD106, lysozyme, cathepsin C and G, COX2, and α1-antitrypsin were downregulated. Therefore, it is suggested 10 μg/mL PTX-treated cells are tend to differentiate into M1 type macrophages, which are pro-inflammatory type and important for phagocytosis and secretion of pro-inflammatory cytokines and microbicidal molecules to defend against pathogens, such as bacteria, virus [60–62], etc. This activation of M1 type macrophage polarization after 10 μg/mL PTX treatment was correlated with the increase of FAS-mediated apoptosis contrary to p53-mediated apoptosis in RAW 264.7 cells.

On the other hand, 300 μg/mL PTX consistently decreased the expression of inflammatory proteins, TNFα, IL-6, CD4, CD80, M-CSF, CXCR4, MMP1, and lactoferrin but compensatory increased the expression of anti-inflammatory protein TGF-β1 compared to 10 μg/mL PTX. And 300 μg/mL PTX increased the expression of anti-apoptosis protein, BCL2, but minimally affected the expression of tumor suppressor protein, p53, compared to 10 μg/mL PTX, whereas the apoptosis triggering TNF family protein, FASL, and the initiator caspase, c-caspase 8 were more upregulated by 300 μg/mL PTX than 10 μg/mL PTX. Therefore, it is suggested 300 μg/mL PTX-treated cells showed anti-inflammatory effect and FAS-mediated apoptosis.

The 10 μg/mL PTX-treated RAW 264.7 cells showed crosstalk between activated RAS and NFkB signalings, and were progressed to cytodifferentiation by upregulating some growth factors and SHH/PTCH signaling. Particularly, it is evident that the 10 μg/mL PTX-treated RAW 264.7 cells have potentials of neuromuscular and osteoblastic differentiation, which are able to influence on objective adjacent cells [63–65]. The active neuromuscular and osteoblastic differentiations were also observed in ICC and western blot in this study. On the other hand, regarding the expression of osteogenesis proteins, 300 μg/mL PTX downregulated the osteoblastic differentiation proteins, BMP2, RUNX2, osterix, osteocalcin, osteopontin, and osteonectin, but upregulated the osteoclastic differentiation protein, RANKL, and compensatory increased the expression of OPG compared to 10 μg/mL PTX.

In the present study, 300 μg/mL PTX consistently decreased the expression of proliferation-, RAS/NFkB signaling-, inflammation-, and osteogenesis-related proteins but apoptosis proteins compared to 10 μg/mL PTX, therefore, it is thought that the high dose 300 μg/mL PTX may somehow disturb the protein expressions and give a harmful effect on RAW 264.7 cells, and resulted in the increase of FAS-mediated apoptosis, whereas the low dose 10 μg/mL PTX showed characteristic protein expression of competitive non-selective phosphodiesterase inhibitor, which may be helpful for the investigation of PTX pharmacological effect in human.

Contrary to the neuromuscular and osteoblastic differentiation in PTX-treated RAW 264.7 cells, 10 μg/mL PTX suppressed fibroblastic differentiation and attenuated collagen production by downregulating the fibrosis-inducing proteins, FGF1, FGF2, TGF-β1, CTGF, collagen 3A1, 4, and 5, laminin α5, integrin β1, α1-antitrypsin, and upregulating the fibrosis-inhibiting proteins, plasminogen, CMG2, integrin α2 and α5. On the other hand, 10 μg/mL PTX increased the expression of endothelin-1 having a key role of vascular homeostasis. In the global expression of 10 μg/mL PTX-treated RAW 264.7 cells, the anti-fibrotic protein expression is closely relevant to the reduction of chronic inflammation-associated M2 macrophage polarization by downregulating CD68, CD106, lysozyme, MMP9, and α1-antitrypsin, and the low level of ROS damage and ER stress after 10 μg/mL PTX treatment. Therefore, it is suggested that 10 μg/mL PTX inhibits M2 type macrophage polarization through RAS/NFkB/TNFα signaling [66], and reduces ER stress through eIF2α/eIF2AK3/GADD153/ATF6 signaling, which negatively regulates the growth, TGF-β1, -β2, -β3, FGF1, FGF2, and CTGF, and extracellular matrix-associated proteins, collagen-3A1, -4, -5A, laminin α5, integrin β1, plasminogen, PAI-1, α1-antitrypsin, and elafin, and eventually resulted in anti-fibrotic effect on RAW 264.7 cells.

10 μg/mL PTX reduced the expression of many angiogenic proteins, including angiogenin, VEGF-A, VEGF-D, vWF, FLT-4, LYVE-1, FGF-2, CD1056 (VCAM-1), MMP-2, MMP-10, PAI-1, CD54, and CD56, but upregulated some angiogenic proteins responsible for wound and damage, that is, HIF1α, VEGF-C, VEGFR2, CMG2, plasminogen, endothelin-1, and CD44. These results indicate 10 μg/mL PTX primarily inhibited angiogenesis but secondarily maintained *de novo* angiogenesis for wound healing [67].

In addition, 10 μg/mL PTX was found to increase the potential of oncogenesis by upregulating the oncogenic proteins, CEA, 14-3-3θ, survivin, mucin 4, and YAP1, and downregulating the tumor suppressor proteins, P53, Rb1, BRCA1, BRCA2, NF1, ATM, maspin, and DMBT1. Nevertheless, 10 μg/mL PTX did not increase the expression of DNA repair enzymes, MBD4 and PARP-1, and exogenous stress responsible proteins, JNK1 and JAK2, but showed the overexpression of antioxidant proteins, SOD1 and GSTO1/2, and cell protection proteins, HSP70, sirtuin 6, and leptin, and resulted in the attenuation of ER stress. Therefore, it is suggested that 10 μg/mL PTX does not exert oncogenesis in RAW 264.7 cells, but maintains the cellular homeostasis, even though there appears slight elevation of some oncogenic protein expression.

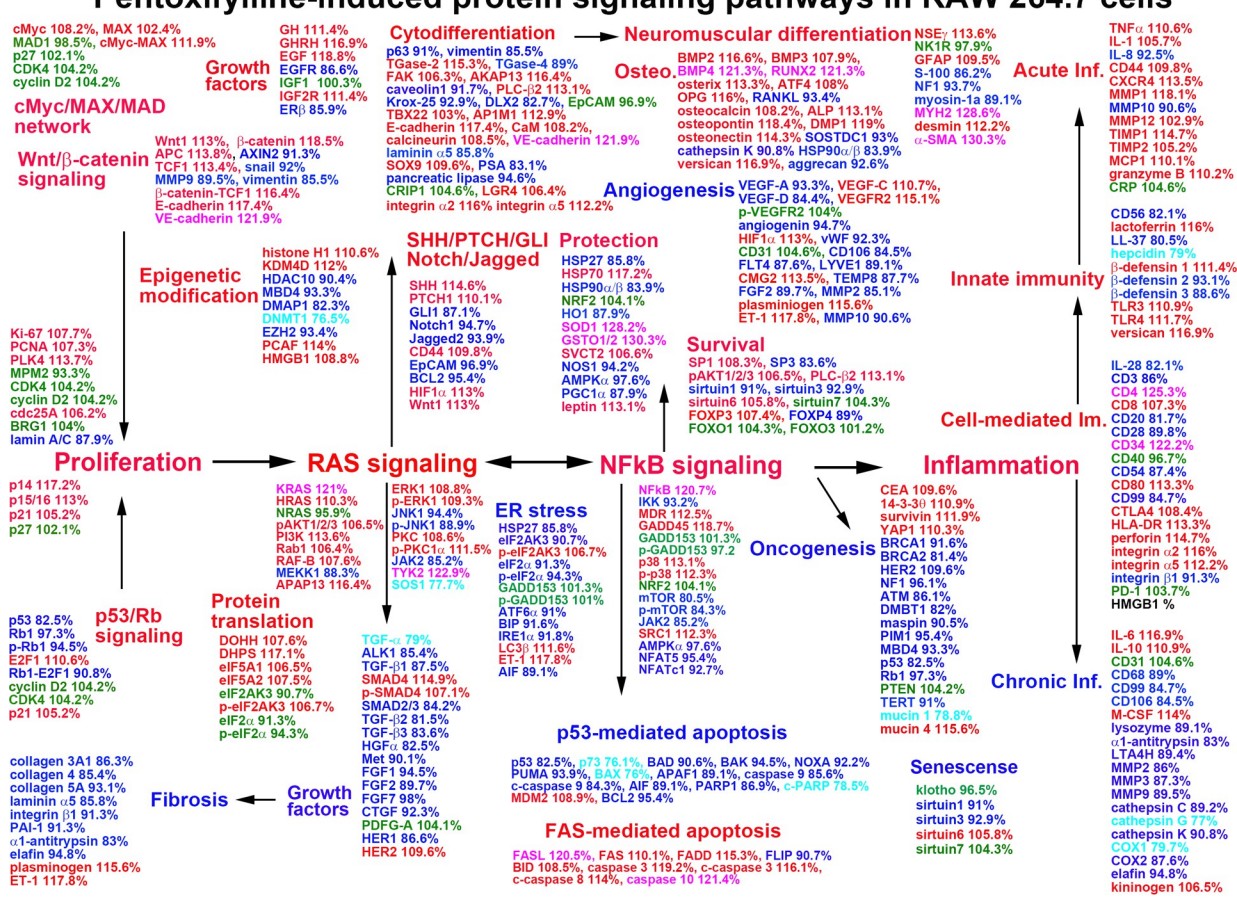

**Fig 19. A diagram of 10 μg/mL PTX-induced protein expression change in global protein signaling pathways of RAW 264.7 cells.** The main axis of cellular signaling, that is, proliferation, RAS signaling, NFkB signaling, protection, survival and aging, and inflammation were consistently activated by 10 μg/mL PTX, and subsequently followed by activation of epigenetic modification, protein translation, Wnt/β-catenin, cMyc/MAX/ MAD network, neuromuscular and osteoblastic differentiation, acute inflammation, innate immunity, cell-mediated immunity, and FAS-mediated apoptosis, while inactivation of p53/Rb/E2F signaling, ER stress, angiogenesis, fibrosis, and chronic inflammation. Sky blue letter: downregulated expression (under 80%), Blue letter: downregulated expression (80–95%), Green letter: minimal change expression (95–105%), Red letter: upregulated expression (105–120%), Flower pink letter: upregulated expression (over 120%).

The 10 μg/mL PTX-induced protein expression changes of different signaling pathways were summarized in a diagram of Fig 19. The panoramic protein signaling diagram illustrated main axes of protein signaling pathways in cells based on IP-HPLC data obtained in this study, therefore, it may indicate the real status of pharmacological effect of PTX in RAW 264.7 cells. We thought this PTX-induced protein expression of different signaling pathways should be corrected or added by further precise protein expression investigation using different cells and animals.

## Conclusions

PTX as a non-selective phosphodiesterase inhibitor showed different biological effect on RWA 264.7 cells depending on the concentration of low dose 10 μg/mL or high dose 300 μg/mL PTX. 10 μg/mL PTX enhanced a central protein expression pathways, RAS/NFkB signaling, and subsequently induced proliferation, epigenetic activation, protection, survival,

neuromuscular and osteoblastic differentiation, and stimulated acute inflammation, innate immunity, and cell mediated immunity, while reduced chronic inflammation, ER stress, and fibrosis but reactively increased FAS-mediated apoptosis and oncogenic potential in RAW 264.7 cells. On the other hand, 300 μg/mL PTX was found to decrease RAS/NFkB signaling compared to 10 μg/mL PTX, and subsequently attenuated proliferation, epigenetic activation, inflammation, neuromuscular and osteoblastic differentiation but increased apoptosis and fibrosis.

## Supporting information

**S1 Fig. Mathematical algorithm for IP-HPLC analysis.**
(PDF)

**S2 Fig. Representative chromatography through IP-HPLC analysis.**
(PDF)

**S1 File. Raw data file of IP-HPLC.**
(XLSX)

**S1 Raw images.**
(DOCX)

## Acknowledgments

We express our gratitude to the late Professor Je Geun Chi who encouraged us to perform IP-HPLC, and to the late Dr. Soo Il Chung who taught us the biological usefulness of IP-HPLC.

## Author Contributions

**Conceptualization:** Mi Hyun Seo, Yeon Sook Kim, Suk Keun Lee.

**Data curation:** Mi Hyun Seo, Dae Won Kim.

**Investigation:** Dae Won Kim, Yeon Sook Kim, Suk Keun Lee.

**Methodology:** Mi Hyun Seo, Dae Won Kim.

**Project administration:** Yeon Sook Kim, Suk Keun Lee.

**Software:** Dae Won Kim, Yeon Sook Kim.

**Supervision:** Suk Keun Lee.

**Validation:** Yeon Sook Kim, Suk Keun Lee.

**Writing – original draft:** Mi Hyun Seo.

**Writing – review & editing:** Yeon Sook Kim, Suk Keun Lee.

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
