## [Decision Letter · Decision Letter 0]

23 Feb 2022

PONE-D-21-38689Pentoxifylline-induced Protein Expression Change in RAW 264.7 Cells as Determined by Immunoprecipitation-based High Performance Liquid ChromatographyPLOS ONE

Dear Dr. Lee,

Thank you for submitting your manuscript to PLOS ONE. After careful consideration, we feel that it has merit but does not fully meet PLOS ONE’s publication criteria as it currently stands. Therefore, we invite you to submit a revised version of the manuscript that addresses the points raised during the review process.

We look forward to receiving your revised manuscript.

Kind regards,

Michael Schubert

Academic Editor

PLOS ONE

“NO”

5. Please check the correct figures 16 in pages 40 and 51.

Reviewers' comments:

Reviewer's Responses to Questions

**Comments to the Author**

1. Is the manuscript technically sound, and do the data support the conclusions?

Reviewer #1: Yes

Reviewer #2: Yes

2. Has the statistical analysis been performed appropriately and rigorously? 

Reviewer #1: N/A

Reviewer #2: Yes

3. Have the authors made all data underlying the findings in their manuscript fully available?

Reviewer #1: Yes

Reviewer #2: Yes

4. Is the manuscript presented in an intelligible fashion and written in standard English?

Reviewer #1: Yes

Reviewer #2: Yes

5. Review Comments to the Author

Reviewer #1: The "unbiased" large scale analysis of multiple signaling components at protein level adds new details to the mode of action, especially of the dose-dependent actions, of this old, but still used drug.

The weak point of the paper is that the results are based on a single mouse macrophage cell line. Since the agent is nowadays rarely used in mouse models, the considerable effort of the authors might be better spent using human models, better mimicking the clinical situation. Although analogous mature human macrophage cell lines are not available, the experiments might at least be supplemented some experiments with primary human macrophages or in vitro "matured" human monocyte cell lines.

The paper is still acceptable in its present form, however, the use of human primary cells or human cell lines would greatly increase its practical value.

Reviewer #2: The present work suggested that PTX has different biological effects on RWA 264.7 cells depending on the concentration of 10 μg/mL and 300 μg/mL PTX. The low dose 10 μg/mL PTX enhanced RAS/NFkB signaling, proliferation, differentiation, and inflammation, particularly, it stimulated neuromuscular and osteoblastic differentiation, innate immunity, and cell-mediated immunity, but attenuated ER stress, fibrosis, angiogenesis, and chronic inflammation, 44 while the high dose 300 μg/mL PTX was found to alleviate the 10 μg/mL PTX-induced biological effects, resulted in the suppression of RAS/NFkB signaling, proliferation, neuromuscular and osteoblastic differentiation, and inflammation. However, I have one comment to the authors as following:

- Could the authors explains what is the difference between this work and the recently published paper "Appl. Sci. 2021, 11, 8273. ?" ext-link-type="uri" xlink:type="simple">https://doi.org/10.3390/app11178273"?

6. PLOS authors have the option to publish the peer review history of their article (what does this mean?). If published, this will include your full peer review and any attached files.

Reviewer #1: No

Reviewer #2: No

---

## [Author Response · Author response to Decision Letter 0]

3 Mar 2022

PLOS ONE Decision: Revision required [PONE-D-21-38689] - [EMID:ca32f9eaaa310396] 

Please, find a file named “Response to Reviewers-220302.docx”.

Please, find a file named “Revised Manuscript with Track Changes-PTX-220302.docx”.

Please, find a file named “ “.

If applicable, we recommend that you deposit your laboratory protocols in protocols.io to enhance the reproducibility of your results. Protocols.io assigns your protocol its own identifier (DOI) so that it can be cited independently in the future. For instructions see: https://journals.plos.org/plosone/s/submission-guidelines#loc-laboratory-protocols. Additionally, PLOS ONE offers an option for publishing peer-reviewed Lab Protocol articles, which describe protocols hosted on protocols.io. Read more information on sharing protocols at https://plos.org/protocols?utm_medium=editorial-emailutm_source=authorlettersutm_campaign=protocols.

[Answer]

Because the experimental protocols used in this experiment are generally simple and easy, it seems that no additional protocol is required. 

We look forward to receiving your revised manuscript.

Kind regards,

Michael Schubert

Academic Editor

PLOS ONE

[Answer]

The present manuscript has been readjusted to the fomatting sample file as indicated.

[Answer]

 The raw western blot images are in Supporting Information. The western blot images of the present experiment were directly obtained from digital image analyzer of ChemiDoc XRS system (Bio-Rad Laboratories, Hercules, CA, USA), which were not followed by any image modification. 

[Answer]

Unfortunately, we have no repository accession number in use. Please, delete my previous statement of repository information. 

“NO”

5. Please check the correct figures 16 in pages 40 and 51.

[Answer]

The authors received no specific funding for this work.

It was corrected, “Fig 19.” in page 52. 

[Answer]

No cited paper was retracted in this manuscript.

Reviewers' comments:

Reviewer's Responses to Questions

Comments to the Author

1. Is the manuscript technically sound, and do the data support the conclusions?

Reviewer #1: Yes

Reviewer #2: Yes

2. Has the statistical analysis been performed appropriately and rigorously?

Reviewer #1: N/A

Reviewer #2: Yes

3. Have the authors made all data underlying the findings in their manuscript fully available?

Reviewer #1: Yes

Reviewer #2: Yes

4. Is the manuscript presented in an intelligible fashion and written in standard English?

Reviewer #1: Yes

Reviewer #2: Yes

5. Review Comments to the Author

Reviewer #1: The "unbiased" large scale analysis of multiple signaling components at protein level adds new details to the mode of action, especially of the dose-dependent actions, of this old, but still used drug.

The weak point of the paper is that the results are based on a single mouse macrophage cell line. Since the agent is nowadays rarely used in mouse models, the considerable effort of the authors might be better spent using human models, better mimicking the clinical situation. Although analogous mature human macrophage cell lines are not available, the experiments might at least be supplemented some experiments with primary human macrophages or in vitro "matured" human monocyte cell lines.

The paper is still acceptable in its present form, however, the use of human primary cells or human cell lines would greatly increase its practical value.

[Answer]

 PTX is an old and well-known drug, but its effect of non-selective phosphodiesterase inhibitor may give an important roles in the treatment of different diseases. Recently, PTX is frequently recommended as a wound healing drug by inhibiting fibrosis and stimulating osteogenesis to prevent severe complications of osteoradionecrosis in head and neck cancer patients. 

 Regarding the selection of human cell line, I am appreciate for your comment using primary human macrophages or in vitro "matured" human monocyte cell lines. It is really critical to use human therapeutic dose of PTX in human cell line, therefore, in the following experiment, we have a plan to perform by using human macrophages or Human Umbilical Vein Endothelial Cells (HUVECs). It is sorry to inform that we are unable to fulfill the reviewer’s comment at this time. 

Reviewer #2: The present work suggested that PTX has different biological effects on RWA 264.7 cells depending on the concentration of 10 μg/mL and 300 μg/mL PTX. The low dose 10 μg/mL PTX enhanced RAS/NFkB signaling, proliferation, differentiation, and inflammation, particularly, it stimulated neuromuscular and osteoblastic differentiation, innate immunity, and cell-mediated immunity, but attenuated ER stress, fibrosis, angiogenesis, and chronic inflammation, 44 while the high dose 300 μg/mL PTX was found to alleviate the 10 μg/mL PTX-induced biological effects, resulted in the suppression of RAS/NFkB signaling, proliferation, neuromuscular and osteoblastic differentiation, and inflammation. However, I have one comment to the authors as following:

- Could the authors explains what is the difference between this work and the recently published paper "Appl. Sci. 2021, 11, 8273. https://doi.org/10.3390/app11178273"?

[Answer]

 Dr. Kim, Soung Min, who is a corresponding author of the paper published in Appl. Sci. 2021, was cooperated with me for the PTX effect on RAW 264.7 cells. After the preliminary experiment, I found a great discrepancy between human therapeutic dose (10 μg/mL) of PTX and high dose (100 – 500 μg/mL) frequently used in many animal experiments. And then, Dr. Kim submitted the preliminary data to the Journal of Applied Science by inclining to the results reported in the literature in the absence of any discussion, therefore, I had retrieved my authorship from the paper.

 The present manuscript is a revised and upgraded one through careful analysis using different methods. We hope this manuscript contribute a little advance in the elucidation of PTX pharmacological effect. 

6. PLOS authors have the option to publish the peer review history of their article (what does this mean?). If published, this will include your full peer review and any attached files.

Do you want your identity to be public for this peer review? For information about this choice, including consent withdrawal, please see our Privacy Policy.

Reviewer #1: No

Reviewer #2: No

[Answer]

All figure files have been examined and converted by PACE digital diagnostic tool.

Besides the editor and reviewer’s comments, the present manuscript is added more IP-HPLC data using 6 antisera, including AXIN2, HMGB1, LGR4. Klotho, FOXO1, and FOXO3. Because the graph of protection proteins become much crowded and complicated, it is divided into two graphs of protection-related proteins and survival and aging-related proteins in Fig 13. And the diagram of global protein signaling pathway of PTX, Fig 19, is upgraded to demonstrate the panoramic scope of protein expression. 

The additional minor correction of mistyped letter or sentences is done and traceable in “Revised Manuscript with Track Changes-PTX-220302.docx”.

Thank you very much for your kindness.

---

## [Editor Report · Decision Letter 1]

7 Mar 2022

Pentoxifylline-induced Protein Expression Change in RAW 264.7 Cells as Determined by Immunoprecipitation-based High Performance Liquid Chromatography

PONE-D-21-38689R1

Dear Dr. Lee,

We’re pleased to inform you that your manuscript has been judged scientifically suitable for publication and will be formally accepted for publication once it meets all outstanding technical requirements.

Kind regards,

Michael Schubert

Academic Editor

PLOS ONE

---

## [Editor Report · Acceptance letter]

11 Mar 2022

PONE-D-21-38689R1 

Pentoxifylline-induced Protein Expression Change in RAW 264.7 Cells as Determined by Immunoprecipitation-based High Performance Liquid Chromatography 

Dear Dr. Lee:

I'm pleased to inform you that your manuscript has been deemed suitable for publication in PLOS ONE. Congratulations! Your manuscript is now with our production department. 

Kind regards, 

on behalf of

Dr. Michael Schubert 

Academic Editor

PLOS ONE